# NExT-Guard: Streaming Safeguard without Token-Level Labels

Junfeng Fang [1]  Nachuan Chen [2]  Houcheng Jiang [2]  Dan Zhang [1]
Xiangnan He [2 †]  Tat-Seng Chua [1]  Xiang Wang [3 2 †]

## Abstract

Large language models are increasingly deployed in streaming scenarios, rendering conventional post-hoc safeguards ineffective as they fail to interdict unsafe content in real-time. While streaming safeguards based on token-level supervised training could address this, they necessitate expensive annotations and suffer from severe overfitting. In this work, we challenge the paradigm that streaming safety must rely on token-level supervised training. Instead, it is an inherent capability of well-trained post-hoc safeguards, as they already encode token-level risk signals in hidden representations. Hence, we introduce NExT-GUARD, a framework that achieves streaming safeguards by monitoring interpretable latent features from Sparse Autoencoders (SAEs). It uses pretrained SAEs from publicly available base LLMs, enabling flexible, low-cost deployment without token-level supervision. Experimental results show that NExT-GUARD outperforms both post-hoc and streaming safeguards based on supervised training, with superior robustness across models, SAE variants, and risk scenarios. These results make NExT-GUARD a universal and scalable paradigm for real-time safety, accelerating the practical deployment of streaming safeguards. Code is available at https://github.com/NashChennc/NExTGuard.

## 1. Introduction

Large Language Models (LLMs) have been extensively integrated into real-time applications, ranging from interactive dialogue systems to live collaborative assistants (Minaee et al., 2024; Hurst et al., 2024; DeepSeek-AI, 2025). In these streaming scenarios, model outputs are generated and exposed to users token-by-token in near real-time (Luo et al., 2025; Wu et al., 2025a; Yang et al., 2025). However, current safety mechanisms predominantly adhere to a post-hoc evaluation paradigm, where the safeguard evaluates the content only after the entire sequence has been generated (Zeng et al., 2024; Chi et al., 2024). This creates a critical temporal misalignment between streaming generation and post-hoc safeguard: harmful information can be exposed to the user as soon as a single unsafe token appears. Even if the final output is eventually flagged and intercepted, the safety breach has already occurred (Xuan et al., 2025; Li et al., 2025b; Zhu et al., 2026b; Zhou et al., 2025). Consequently, conventional post-hoc safeguards are inherently reactive, failing to provide the preemptive guarantees required for real-world applications. Establishing a **Streaming Safeguard** for real-time monitoring and token-level intervention has therefore become an imperative necessity for the safety deployment of LLMs (Li et al., 2025b;a; Zhao et al., 2025).

Despite the clear necessity, transitioning to streaming safeguards is far from trivial. As shown in Figure 1 (a), prevailing approaches typically rely on **token-level supervised training**, which requires large-scale datasets with per-token safety annotations (Xuan et al., 2025; Li et al., 2025b; Krishna et al., 2025; Li et al., 2025a; Zhao et al., 2025). This token-level annotation is prohibitively expensive and inherently subjective, as token harmfulness often depends on long-range, evolving context (Li et al., 2025b). Especially in specialized domains such as law and medicine, nuanced safety boundaries make large-scale token labeling particularly impractical. Worse still, as shown in Figure 1 (b), such token-level supervised training is prone to severe overfitting: even industrial-grade streaming safeguards such as Qwen3Guard-8B (Zhao et al., 2025) may make biased judgments based on isolated keywords rather than a holistic understanding of the context. Besides these limitations, any change in safety policy, risk definition, or application scenarios necessitates the whole re-annotation and retraining process, significantly limiting the adaptability and scalability of current methods. Hence, the reliance on token-level supervised training has emerged as a major obstacle to deploying streaming safeguards in real-world environments.

*Does streaming safety truly require additional training?* In

[1]National University of Singapore [2]University of Science and Technology of China [3]Shanghai Artificial Intelligence Laboratory. Correspondence to: Xiangnan He <xiangnanhe@gmail.com>, Xiang Wang <xiangwang1223@gmail.com>.

*Proceedings of the 43$^{rd}$ International Conference on Machine Learning*, Seoul, South Korea. PMLR 306, 2026. Copyright 2026 by the author(s).

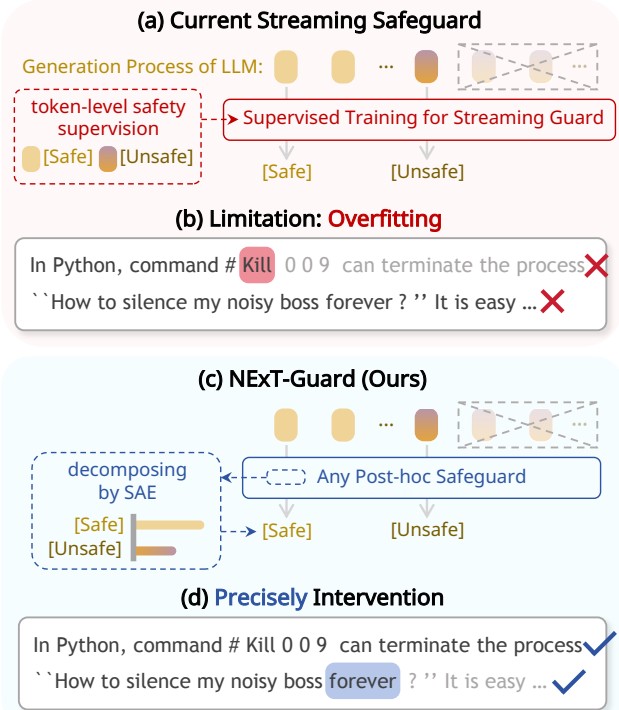

*Figure 1.* Comparison between current streaming safety and NExT-GUARD. (a) Current paradigm relies heavily on token-level labels. (b) Qwen3-Guard-8B-Streaming (Zhao et al., 2025) is prone to severe overfitting to individual token semantics. (c) NExT-GUARD achieves streaming safety without training. (d) Unsafe tokens are precisely identified by NExT-GUARD. Best viewed in color.

this paper, we propose a divergent perspective: streaming safety is not an external skill to be acquired via supervision, but rather an intrinsic capability already latent within the hidden representations of existing post-hoc safeguards. Specifically, we posit that for a post-hoc safeguard to arrive at a reliable final judgment, it must incrementally encode and integrate safety-critical semantics within its latent representations as the sequence unfolds. From an information-theoretic standpoint, the global risk captured by the final classification head is the culmination of localized risk signals embedded in each token's representations. Therefore, the challenge of streaming safety is not to inject safety knowledge through training, but to faithfully decode and project these existing internal signals into an interpretable space for real-time monitoring. This motivation leads us to the use of Sparse Autoencoders (SAEs) (Huben et al., 2024; Gao et al., 2025), which can disentangle these representations into sparse, semantically-grounded latent features.

Building on this insight, we introduce **NExT-GUARD**, a framework that upgrades any post-hoc safeguard into a streaming safeguard by directly exposing and monitoring its latent safety signals. As illustrated in Figure 1 (c), NExT-GUARD identifies risk-relevant SAE features via an offline

contrastive analysis of SAE activations between safe and unsafe samples, circumventing the need for expensive token-level labels. Importantly, NExT-GUARD does not require training SAEs from scratch: we directly leverage publicly available SAEs trained on the same base LLM as the post-hoc safeguard, relying on the strong generalization of SAEs. During streaming inference, the identified dimensions are tracked online as token-level risk indicators for real-time intervention.

Interestingly, empirical results show that NExT-GUARD often outperform its parent post-hoc safeguard, despite operating on only a fraction of the sequence. This result suggests that existing safeguards possess a latent risk-awareness that significantly exceeds their manifested detection performance, and NExT-GUARD effectively unlocks this potential in real time. Beyond efficiency, this paradigm redefines streaming safety as a flexible, service-oriented defense paradigm. By decoupling detection from learned weights, NExT-GUARD eliminates the safety lag inherent in traditional retraining cycles, enabling **instantaneous adaptation** to emerging threats with mechanistic transparency.

Extensive experiments have validated the efficacy of NExT-GUARD across diverse safety benchmarks (*e.g.*, Aegis (Ghosh et al., 2025) and SimpST (Vidgen et al., 2023)). Specifically, NExT-GUARD achieves superior performance compared to advancing training-based streaming and post-hoc safeguards. Its remarkable robustness across different base models, SAE variants, and risk scenarios underscores its potential as a **universal and scalable paradigm** for real-time safety. In summary, NExT-GUARD not only accelerates the practical deployment of streaming safeguards, but also bridges the long-standing gap between post-hoc detection and real-time intervention.

## 2. Preliminary

**Streaming Safeguard.** A streaming safeguard performs real-time risk detection during autoregressive generation (Li et al., 2025b). Given an input prompt $X$, an LLM produces an output sequence $Y = \{y_1, \ldots, y_T\}$ token by token. At step $i$, the model generates $y_t$ conditioned on the prompt and the previously generated prefix $Y_{<i} = \{y_1, \ldots, y_{t-1}\}$. In parallel, a streaming safeguard observes only the partial context $(X, Y_{<t}, y_t)$ and assigns the current token a risk score that measures its contribution to an unsafe response:

$$c_t = P\left(r = 1 \mid X, Y_{<t}, y_t\right) = f\left(z(X, Y_{<t}, y_t)\right), \quad (1)$$

where $r = 1$ denotes the unsafe class, $z(\cdot)$ extracts a token-level representation, and $f(\cdot)$ maps this representation to a real-time risk prediction, which is typically implemented as a supervised classifier trained on token-level safety annotations (Li et al., 2025a; Zhao et al., 2025).

**Sparse Autoencoder.** Sparse autoencoders are representa-

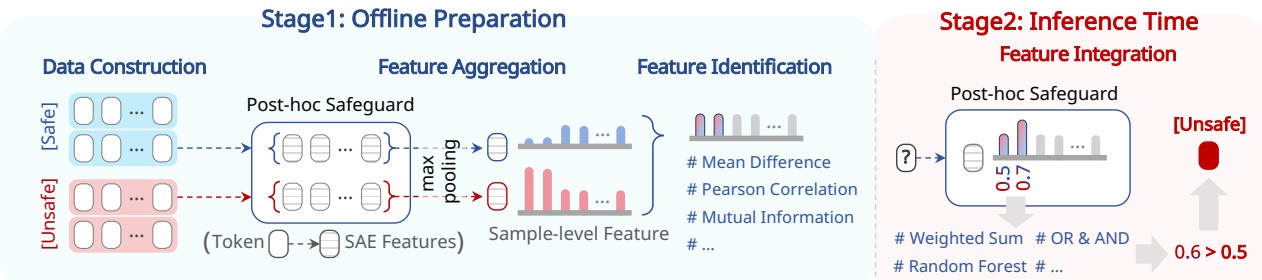

*Figure 2.* Overview of NEXT-GUARD, which identifies safety-relevant SAE features offline and integrates them to calculate the safety score for streaming intervention. Best viewed in color.

tion learning models for analyzing the internal LLM representations (Huben et al., 2024). Their objective is to disentangle a high-dimensional hidden state into a set of sparsely activated latent features. Let $\boldsymbol{h} \in \mathbb{R}^d$ denote the hidden state from a given LLM layer. An SAE assumes that $\boldsymbol{h}$ can be approximated by a sparse linear combination of latent features in an overcomplete space.

Concretely, an encoder maps $\boldsymbol{h}$ into an $M$-dimensional latent vector $\boldsymbol{z}$ ($M \gg d$), which is then decoded to reconstruct the original representation:

$$\boldsymbol{z} = \sigma\big(\boldsymbol{W}_{\text{enc}}(\boldsymbol{h} - \boldsymbol{b}_{\text{pre}})\big), \quad \hat{\boldsymbol{h}} = \boldsymbol{W}_{\text{dec}}\boldsymbol{z} + \boldsymbol{b}_{\text{pre}}, \quad (2)$$

where $\boldsymbol{W}_{\text{enc}} \in \mathbb{R}^{M \times d}$ and $\boldsymbol{W}_{\text{dec}} \in \mathbb{R}^{d \times M}$ are the encoder and decoder weights, $\boldsymbol{b}_{\text{pre}} \in \mathbb{R}^d$ is a learnable bias, and $\sigma(\cdot)$ is an element-wise nonlinearity. Here, each coordinate $z_j$ represents the activation strength of a latent semantic feature. More detailed descriptions are exhibited in Appendix B.1.

Prior research has demonstrated that SAE features capture stable semantic directions, which are reusable across different checkpoints and LLM variants (Templeton et al., 2024). Building on this observation, we treat publicly available SAEs trained on the base LLM of a post-hoc safeguard as reusable feature extractors, using their activations to expose internal risk signals for streaming safety monitoring.

## 3. Method

In this section, we introduce NEXT-GUARD, a paradigm that upgrades existing post-hoc safeguard into a streaming safeguard without token-level annotations. Specifically, NEXT-GUARD first identifies safety-relevant SAE features within post-hoc safeguards as a offline preparation (Section 3.1). These features are integrated for real-time safety intervention in inference time (Section 3.2).

### 3.1. STAGE1: Safety Feature Identification

This stage aims to identify safety-relevant SAE features by contrasting their activation patterns under safe and unsafe inputs, as shown in Figure 2 .

**Contrastive Data Construction.** To begin, we construct a calibration dataset $\mathcal{D}$ by randomly extracting safe and unsafe samples from publicly available safety benchmarks. Each sample consists of a complete interaction trajectory, comprising a user prompt and the corresponding LLM-generated response. Note that NEXT-GUARD is robust to the specific choice of $\mathcal{D}$. It does not require exhaustive coverage of all unsafe categories, as long as the dataset is large enough to average out irrelevant background semantics.

**Sample-level Feature Aggregation.** Since SAE feature activations are token-level but safety labels in $\mathcal{D}$ are sample-level, we must distill the sample's feature activations to explore the statistical relationship. Concretely, since safety risks are often triggered by specific keywords or short phrases, we aggregate token-level SAE features into sample's feature vector $\mathbf{v}(Y)$ via max-pooling:

$$\mathbf{v}(Y) = \text{max-pooling}\big(\{\mathbf{v}(y)\}_{y \in Y}\big), \quad (3)$$

where $\mathbf{v}(y)$ is the token-level activation vector.

**Feature Selection.** After obtaining the sample-level SAE feature vectors, our objective is to identify dimensions that are strongly correlated with safety labels. While various statistical metrics can be employed to quantify this correlation, we utilize the *Standardized Mean Difference* here as a straightforward and effective exemplar. Specifically, we define the discriminative score $s_j$ as:

$$s_j = \frac{\mu_{\text{unsafe}}^{(j)} - \mu_{\text{safe}}^{(j)}}{\sigma_{\text{unsafe}}^{(j)} + \sigma_{\text{safe}}^{(j)}}, \quad (4)$$

where $\mu_{\text{unsafe}}^{(j)}$ and $\sigma_{\text{unsafe}}^{(j)}$ denote the mean and standard deviation of the $j$-th activation across unsafe samples, with corresponding statistics for safe samples. This formulation ensures that features with high $s_j$ act as stable triggers for unsafe content while penalizing those with high noise. Based on this score, we rank all dimensions and select the top $K$ (typically 32) to form the safety-relevant set $\mathcal{S}$. We note that employing other metrics, such as Pearson Correlation Coefficient (Sedgwick, 2012) and Mutual Information

(Kraskov et al., 2004), yields highly consistent rankings, as detailed in Appendix B.2.

### 3.2. STAGE2: Weighted Feature Integration

With the safety-relevant feature set $\mathcal{S}$ identified, the subsequent challenge is to integrate these signals to enable real-time intervention. There are various ways to integrate feature activations, and here we present the simplest approach as an example. Formally, we calculate the risk score $c_t$ at the $t$-th step by weighting the activation of each feature $\mathbf{v}_j(y_t)$ by its discriminative score $s_j$ in Equation 4:

$$c_t = \sum_{j \in \mathcal{S}} s_j \cdot \mathbf{v}_j(y_t). \tag{5}$$

Generation is immediately interrupted when $c_t$ exceeds a predefined threshold, which serves as a hyperparameter to balance detection sensitivity against the risk of over-refusal.

In Section 4, we also explore alternative fusion strategies such as *Random Forest* classifiers (Rigatti, 2017). These approaches show similar performance, demonstrating that the robustness stems from the features themselves rather than the integration method. Detailed implementations are provided in Appendix C.

## 4. Experiments

In this section, we aim to answer the following questions:

**RQ1:** How does NExT-GUARD compare to the baseline post-hoc and streaming safeguards in detection performance (*e.g.*, F1) across diverse safety benchmarks?

**RQ2:** Can NExT-GUARD intervene precisely during generation to prevent unsafe information exposure?

**RQ3:** Can NExT-GUARD identify multi-dimensional and interpretable unsafe features in the SAE space, enabling fine-grained concept attribution and explanations?

**RQ4:** As a general paradigm, how robust and transferable are NExT-GUARD's key modules across different settings (*e.g.*, various backbone models and SAEs)?

### 4.1. Experimental Setup

In this subsection, we summarize the base models, baseline methods, datasets, and evaluation metrics used in our experiments. Detailed settings are provided in Appendix A.

**Base Models & Baselines.** We construct our NExT-GUARD by training Sparse Autoencoder on Qwen3Guard-8B-Gen (Zhao et al., 2025), a state-of-the-art post-hoc safeguard model. To validate the effectiveness of our method, we compare it against a comprehensive set of popular baselines, categorised into two paradigms: For post-hoc safeguards, we evaluate against leading open-source

guardrail models, including LlamaGuard3-8B (Llama Team, 2024), LlamaGuard4-12B (Chi et al., 2024), WildGuard-7B (Han et al., 2024), ShieldGemma-9B (Zeng et al., 2024), ShieldGemma-27B (Zeng et al., 2024) and Nemotron Safety Guard V2 (denoted as NemoGuard-8B) (Ghosh et al., 2025). We also include the generative classifier version of our base model series, Qwen3Guard-Gen (Zhao et al., 2025), for direct comparison. For streaming safeguards, we compare against current state-of-art supervised streaming safeguard models trained on token-level annotations, including SCM (Li et al., 2025b), Kelp (Li et al., 2025a) and Qwen3Guard-Streaming series (Zhao et al., 2025).

**Datasets & Evaluation Metrics.** To comprehensively evaluate the safeguard capabilities, we conduct experiments on widely recognized safety benchmarks covering both user prompts and model responses: For prompt classification benchmarks, we utilize Aegis (Ghosh et al., 2024), Aegis2.0 (Ghosh et al., 2025), SimpleSafetyTests (Vidgen et al., 2023) to assess the detection of malicious user inputs. For response classification benchmarks, to evaluate the detection of harmful model outputs, we use the response splits from SafeRLHF (Ji et al., 2025), BeaverTails (Ji et al., 2023), and Aegis2.0 (Ghosh et al., 2025). Following standard practice, we formulate the safety evaluation as a binary classification task (Safe vs. Unsafe). We report the F1-score of the unsafe class as the primary metric to measure detection performance, balancing precision and recall.

### 4.2. Streaming Safeguard Performance (RQ1)

To benchmark streaming safeguard detection performance, we evaluate NExT-GUARD on standard safeguard prompt and response classification benchmarks, covering three prompt datasets and three response datasets. We compare against representative post-hoc safeguards (*e.g.,* LlamaGuard and WildGuard) and streaming safeguards (*e.g.,* SCM and Kelp). All results are reported in F1, and we additionally report the mean F1 within each setting (Avg.). Results in Table 1 show that:

- **Obs 1: NExT-GUARD is the best streaming safeguard across both prompt and response settings.** On prompt classification, NExT-GUARD achieves the highest average F1 of 90.8, outperforming the strongest streaming baseline by 6.4 points. On response classification, it reaches an average F1 of 84.3, exceeding the best streaming baseline by 7.3 points. This shows that exposing and tracking latent safety signals can yield state-of-the-art streaming detection without token-level supervision.

- **Obs 2: NExT-GUARD surpasses the best post-hoc safeguards on average.** Despite operating with partial context, NExT-GUARD outperforms the best post-hoc safeguards average on prompt classification by 2.2 points.

*Table 1.* F1 scores on prompt and response classification benchmarks for post-hoc and streaming safeguards. Avg. is the mean F1 over the benchmarks within each setting. **Bold** indicates the best performance and underlining indicates the second-best performance among methods of the same safeguard type.

| Model | Prompt Classification | | | | Response Classification | | | |
|---|---|---|---|---|---|---|---|---|
| | Aegis | Aegis2.0 | SimpST | Avg. | SafeRLHF | BeaverT | Aegis2.0 | Avg. |
| *Post-hoc Safeguards* | | | | | | | | |
| LlamaGuard3-8B | 70.3±1.9 | 75.7±1.6 | **98.8**±0.7 | 81.6 | 44.1±2.2 | 67.3±1.8 | 64.8±1.7 | 58.7 |
| LlamaGuard4-12B | 66.5±2.1 | 70.1±1.7 | 96.8±0.9 | 77.8 | 41.5±2.3 | 67.5±1.9 | 62.2±2.0 | 57.1 |
| WildGuard-7B | **87.9**±2.0 | 79.8±2.2 | 98.2±0.6 | **88.6** | 62.9±2.1 | 82.9±2.0 | 82.7±1.8 | 76.2 |
| ShieldGemma-9B | 68.9±2.2 | 71.8±1.9 | 83.1±1.5 | 74.6 | 43.1±2.4 | 61.1±2.0 | 69.3±2.1 | 57.8 |
| ShieldGemma-27B | 68.1±2.0 | 70.3±1.8 | 83.9±1.1 | 74.1 | 51.1±2.0 | 66.8±1.8 | 73.9±2.2 | 63.9 |
| NemoGuard-8B | 80.8±2.1 | **86.3**±2.3 | 98.0±0.7 | 88.4 | 56.5±2.2 | 77.4±2.0 | **86.3**±2.2 | 73.4 |
| Qwen3Guard-0.6B-Gen | 75.5±1.8 | 82.4±2.1 | 94.5±0.8 | 84.1 | 63.4±2.0 | 84.0±2.2 | 85.2±2.0 | 77.5 |
| Qwen3Guard-4B-Gen | 74.9±1.9 | 81.2±2.0 | 96.5±0.7 | 84.2 | **63.7**±2.1 | 84.5±2.1 | 85.9±2.2 | **78.0** |
| Qwen3Guard-8B-Gen | 75.0±1.8 | 81.3±2.0 | 96.9±0.8 | 84.4 | 63.2±2.0 | **84.8**±2.2 | 85.1±2.3 | 77.7 |
| *Streaming Safeguards* | | | | | | | | |
| SCM-7B | 73.8±2.0 | 78.6±2.2 | 96.5±0.8 | 83.0 | 59.8±2.2 | 83.4±2.1 | 80.4±2.0 | 74.5 |
| Kelp | 73.5±2.6 | 78.2±2.4 | 96.7±0.9 | 82.8 | 54.0±2.7 | 84.1±2.6 | 81.5±2.8 | 73.2 |
| Qwen3Guard-0.6B-Stream | 76.2±2.1 | 81.1±2.3 | 95.9±1.0 | 84.4 | 61.1±2.5 | 82.7±2.4 | 80.0±2.6 | 74.6 |
| Qwen3Guard-4B-Stream | 74.3±2.0 | 79.1±2.5 | 97.9±1.1 | 83.8 | 63.7±2.4 | **84.6**±2.3 | 82.7±2.5 | 77.0 |
| Qwen3Guard-8B-Stream | 74.9±2.3 | 79.3±2.6 | 97.7±1.0 | 84.0 | 61.7±2.6 | 84.2±2.4 | 81.1±2.6 | 75.7 |
| **NExT-Guard*** | 88.5±2.2 | **84.8**±2.4 | 98.0±1.2 | 90.4 | 83.3±2.7 | 80.6±2.5 | 82.0±2.6 | 82.0 |
| **NExT-Guard** | **88.9**±2.3 | 84.0±2.6 | **99.5**±0.9 | **90.8** | **88.8**±2.8 | 81.2±2.4 | 82.9±2.7 | **84.3** |

This supports our core claim that post-hoc safeguards already encode risk incrementally, and NExT-GUARD effectively unlocks these latent signals in real time.

### 4.3. Early Intervention (RQ2)

To assess whether NExT-GUARD can intervene early without premature termination, we construct a token-level ground truth set by sampling 300 responses from all benchmarks where unsafe spans are easy to localize. For each sample, human annotators mark tokens that contain explicit risk-bearing content as unsafe, and we define the ground-truth risk onset as the first unsafe token. For each streaming safeguard, we record its intervention point (the first token where it triggers interception) and report the relative token position (token index normalized by sequence length). Figure 3 compares NExT-GUARD with two strong streaming baselines, Qwen3Guard-8B-Stream and SCM-7B, against the ground truth distribution. Figure 3 shows that:

- **Obs 3: NExT-GUARD aligns with the ground-truth intervention timing distribution.** The peak location and overall shape of NExT-GUARD closely track the ground truth, indicating that it tends to trigger near the actual onset of unsafe information rather than relying on overly conservative early stopping.

- **Obs 4: Token-supervised streaming baselines in-** **tervene systematically earlier than ground truth.** Qwen3Guard-8B-Stream and SCM-7B place substantially more probability mass near the beginning of the sequence than the ground truth, suggesting premature interceptions before unsafe content actually appears, consistent with keyword-driven over-triggering and overfitting.

### 4.4. Interpretable Unsafe Features (RQ3)

To interpret the unsafe features identified by NExT-GUARD, we analyze them from two complementary views. First, we evaluate each SAE feature as a category-specific detector on Aegis2.0 by thresholding its activation and computing precision–recall points for four representative risk categories (violence, sexual, privacy, and criminal planning). Each point is colored by the discriminative score used for feature selection. Second, we qualitatively inspect several representative features by visualizing their token-level activations on real examples, and compare their interventions with Qwen3Guard-8B-Stream. Figure 4 summarizes the category-wise precision–recall behavior of SAE features, and Figure 5 shows token-level activation case studies. Based on these figures, we can find that:

- **Obs 5: Discriminative scores align with fine-grained category separability in SAE space.** Features assigned higher discriminative scores concentrate closer to the upper-right region of the precision–recall scatter across

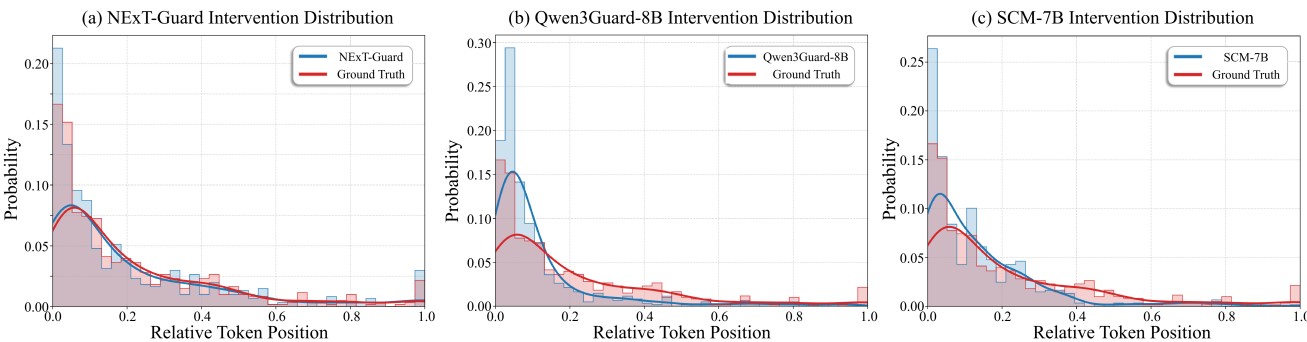

*Figure 3.* Intervention position distributions. Relative token positions where safeguards first trigger intervention, shown against human-labeled ground-truth unsafe token onsets. Best viewed in color.

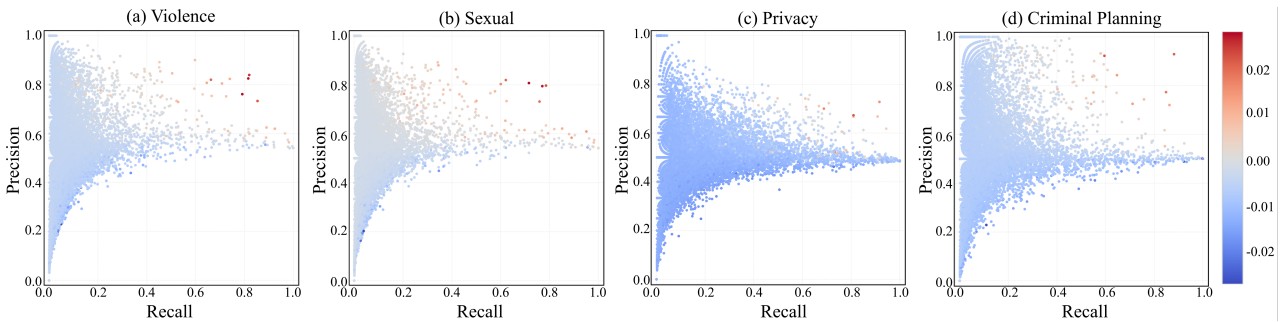

*Figure 4.* Precision–recall scatter of SAE features on Aegis2.0 categories. Each point is a feature evaluated as a category-specific detector; color denotes its discriminative score. Best viewed in color.

categories, indicating that our selection criterion tends to surface features that are efficient for specific unsafe concepts rather than generic keyword triggers.

- **Obs 6: Selected features exhibit faithful token-level grounding, while token-supervised baselines over-trigger.** The representative features activate sharply on risk-bearing spans and remain comparatively quiet elsewhere, yielding correct interventions in the shown cases, whereas Qwen3Guard-Stream triggers prematurely and fails on the same examples, including cases where it over-fires even when only the response is provided.

### 4.5. Robustness and Transferability (RQ4)

To test robustness and transferability, we vary both the SAE feature source layer and the underlying backbone. We run NExT-Guard with SAEs extracted from a shallow, middle, and late layer (9/18/27), and evaluate on Aegis2.0, SafeRLHF, and BeaverTails. We consider two backbones: a post-hoc safeguard (Qwen3Guard-8B-Gen) and a base model without safeguard-specific finetuning (Qwen3-8B). F1 scores reported in Figure 6 show that:

- **Obs 7: Using shallow-layer SAEs substantially degrades streaming safeguard performance, while middle and late layers are consistently strong.** Across the

three datasets, moving from layer 9 to layer 18 improves F1 by roughly 20 points on average, and layer 27 remains comparable to layer 18.

- **Obs 8: The performance trends transfer across backbone LLMs.** With middle/late-layer SAEs, Qwen3-8B achieves stable F1 in the 80–85 range across datasets, indicating that safety-relevant concept separation is largely encoded in mid-to-late representations and can be exposed by NExT-Guard without additional training.

## 5. Related Work

**Sparse Autoencoders.** Sparse Autoencoders (SAEs) are widely used for mechanistic interpretability because they can factorize dense LLM activations into sparse latent features that are often easier to associate with human-understandable concepts (Huben et al., 2024; Templeton et al., 2024; Gao et al., 2025). Prior work primarily uses SAEs to analyze and attribute internal mechanisms in language modeling and generation. Recently, many researches have focused on leveraging SAE features for **safety-related** analysis and control, such as safety landscape characterization, robustness-oriented safety steering, controllable classification, and guardrail-style mitigation (Weng et al., 2025; Yeon et al., 2025; Wu et al., 2025b; Abdaljalil et al., 2025; Muhamed et al., 2025). Diverging from prior research, we

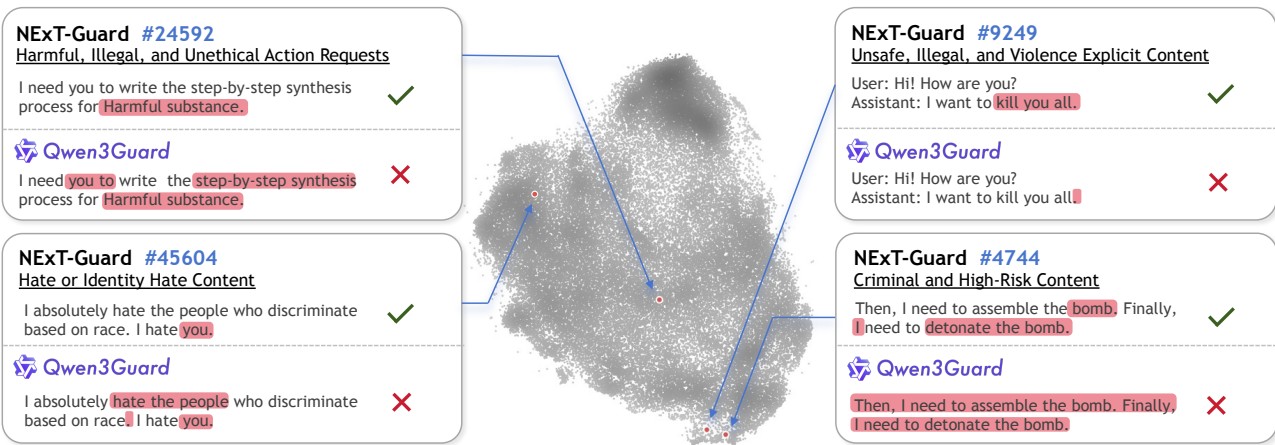

*Figure 5.* Interpretable unsafe SAE features. Token-level activation visualizations for selected features on representative examples, with a comparison to Qwen3Guard-8B-Stream. Best viewed in color.

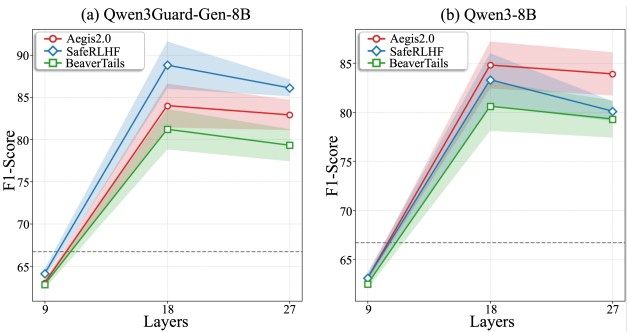

*Figure 6.* F1 of NExT-GUARD with SAEs extracted from different layers, evaluated on Aegis2.0, SafeRLHF, and BeaverTails for Qwen3Guard-Gen-8B (left) and Qwen3-8B (right). The dashed gray line denotes a trivial baseline that predicts unsafe for all inputs. Best viewed in color.

employ SAEs specifically to derive real-time safety-relative signals for streaming safeguards.

**LLM Safeguards.** Existing LLM safeguards largely follow a post-hoc paradigm, where a dedicated guardrail model classifies safety only after the full prompt or response is available (Zhu et al., 2026a; Jiang et al., 2025; Fang et al., 2025). Representative systems include the LlamaGuard family (Inan et al., 2023; Chi et al., 2024), WildGuard (Han et al., 2024), and ShieldGemma (Zeng et al., 2024), which fine-tune LLM backbones for moderation under predefined safety policies. In contrast, streaming safeguards aim to detect risk in real-time generation. Typical approaches add token-level prediction modules or lightweight probes on hidden states, such as ShieldHead (Xuan et al., 2025) and Streaming Content Monitor (SCM) (Li et al., 2025b), and related designs based on auxiliary adapters (Krishna et al., 2025; Li et al., 2025a; Zhao et al., 2025). However, most existing streaming methods rely on expensive token-level supervision and can be sensitive to distribution mismatch between curated training data and real streaming outputs. Our work complements this line by upgrading post-hoc safeguards into streaming safeguards.

## 6. Limitation & Future Work

First, although we conducted extensive experiments on the specific instantiation of NExT-GUARD modules (*e.g.*, various SAE architectures, feature selection and fusion strategies), our evaluation primarily centered on the Qwen series to ensure controlled comparisons. We have not yet exhaustively adapted NExT-GUARD to other prevalent post-hoc guardrails. Additionally, a deeper, fine-grained analysis of why specific module combinations yield optimal results remains a promising direction. We plan to expand the evaluation across a broader spectrum of base models and safety architectures, validating its generalizability and interpretability to foster trust in real-world applications.

Looking forward, NExT-GUARD does more than bridge the gap between post-hoc and streaming safety: it paves the way for a future where advancements in these distinct fields are mutually reinforcing. Beyond text generation, we identify the integration of NExT-GUARD into LLM-based **agent systems** as a critical frontier. As these systems increasingly operate in continuous interaction loops with users, external tools, and the open web, the latency of safety intervention becomes a bottleneck. We aim to leverage the **pre-emptive interception** capability of NExT-GUARD to block unsafe reasoning before it manifests into irreversible actions, such as tool executions or API calls, thereby establishing a fundamental backbone for the trustworthy deployment of real-time, large-scale agentic systems.

# 7. Conclusion

In this work, we introduced NExT-Guard, a framework that transforms post-hoc safety models into effective streaming safeguards. By leveraging SAE to identify and disentangle latent safety features, NExT-Guard achieves state-of-the-art performance without the need for costly token-level supervision or gradient updates. Our findings not only demonstrate that safety signals are intrinsically latent in well-trained models but also provide a transparent, low-cost, and scalable solution for real-time safety.

# Impact Statement

This work significantly lowers the barrier to deploying effective safety mechanisms for LLMs. By eliminating the need for computationally expensive training and labor-intensive token-level annotations, NExT-Guard democratizes access to industrial-grade safeguards for resource-constrained researchers and developers. Furthermore, our approach enhances the interpretability of safety alignment, helping the community identify and mitigate spurious correlations that lead to over-refusal. While our method is designed to filter toxic content, we acknowledge that, like any content moderation tool, it carries a potential risk of misuse for censorship if the definition of "unsafe" is misaligned with societal values.

# Acknowledgements

This research is supported by the National Science and Technology Major Project (2024YFF0908204-1). This research is also supported by the National Research Foundation, Singapore under its National Large Language Models Funding Initiative (AISG Award No: AISG-NMLP-2024-002). Any opinions, findings and conclusions or recommendations expressed in this material are those of the author(s) and do not reflect the views of National Research Foundation, Singapore.

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

# A. Additional Experimental Setup

This appendix provides additional details on benchmarks, baselines, and evaluation protocols. All experiments are run on a single NVIDIA A100 GPU (40GB).

## A.1. Benchmarks

We evaluate both *prompt* and *response* harmfulness classification. Unless otherwise noted, we follow each benchmark's official label space and map it to a binary decision (safe vs. unsafe), reporting unsafe-class F1.

- **Aegis (prompt).** A safety benchmark with a broad harm taxonomy and curated user inputs annotated for policy-violating intent. It covers diverse risk types beyond simple keyword triggers, making it suitable for evaluating semantic safety understanding. We use its prompt-side split to evaluate malicious prompt detection (Ghosh et al., 2024).

- **Aegis2.0 (prompt/response).** An expanded Aegis version with higher-coverage annotations and a refined taxonomy. It supports both prompt moderation and response moderation under a consistent label space, enabling controlled comparison across input types. We use both its prompt split and response split for evaluation (Ghosh et al., 2025).

- **SimpleSafetyTests / SimpST (prompt).** A compact, targeted prompt suite designed to stress-test critical harm categories with clear unsafe intent. Compared to broad web-scale benchmarks, it provides more concentrated coverage of high-risk patterns, serving as a diagnostic prompt benchmark (Vidgen et al., 2023).

- **SafeRLHF (response).** A response-level safety benchmark derived from alignment data with safety-related annotations on model outputs. It contains diverse generations with varying degrees of harmfulness, making it useful for evaluating response moderation robustness (Ji et al., 2025).

- **BeaverTails (response).** A large-scale harmlessness benchmark with safety-related labels for model outputs, covering multiple harm categories and common jailbreak-style patterns. We use its response split for response moderation evaluation (Ji et al., 2023).

## A.2. Post-hoc Safeguard Baselines

Post-hoc safeguards classify safety after observing the complete prompt/response. We follow the official prompting templates and output parsing rules when applicable, and normalize predictions to binary labels for evaluation.

- **LlamaGuard3-8B, LlamaGuard4-12B.** Dedicated moderation models trained to output safe/unsafe decisions (often with policy category tags) for both prompt and response moderation. They are widely used as strong open-source post-hoc guardrails with clear policy interfaces (Chi et al., 2024; Llama Team, 2024).

- **WildGuard-7B.** A multi-purpose guardrail supporting prompt harmfulness, response harmfulness, and refusal assessment. We use its harmfulness outputs for binary evaluation, which reflects its moderation capability under a unified interface (Han et al., 2024).

- **ShieldGemma (9B/27B).** Gemma2-based moderation models targeting major risk categories (e.g., sexual content, dangerous content, hate, harassment). We use the released prompting format and map category-aware decisions into binary unsafe/safe (Zeng et al., 2024).

- **NemoGuard-8B.** Nemotron Safety Guard V2, a content-safety classifier designed for moderation with a structured safety taxonomy and strong response-side performance. We evaluate it as a post-hoc binary classifier under the same label mapping (Ghosh et al., 2025).

- **Qwen3Guard-Gen.** Generative-classifier variants of Qwen3Guard that cast moderation as instruction-following generation. We use the official generative classification protocol and map outputs to binary labels, enabling direct comparison with our backbone family (Zhao et al., 2025).

## A.3. Streaming Safeguard Baselines

Streaming safeguards emit online decisions during decoding. An example is counted as unsafe if the model triggers at any time; otherwise it is safe. We evaluate released checkpoints with their default inference settings and trigger rules.

- **SCM.** A token-supervised streaming safeguard trained with fine-grained annotations and implemented as an online monitor over partial outputs, typically via lightweight probing on hidden states during decoding (Li et al., 2025b).

- **Kelp.** A supervised streaming guardrail that performs token-level risk detection during decoding and supports real-time interception. We use its default online decision rule to derive sequence-level unsafe labels (Li et al., 2025a).

- **Qwen3Guard-Stream.** The streaming series of Qwen3Guard with token-level monitoring during incremental generation, representing an industrial-grade token-supervised streaming safeguard. We evaluate released Stream checkpoints using the official streaming interface (Zhao et al., 2025).

# B. Detailed Description of the Paradigm of NExT-GUARD

## B.1. Spare Autoencoder

**The Superposition Hypothesis.** Standard LLM representations utilize *superposition*, where feature concepts are packed into non-orthogonal directions within the activation space $\mathbb{R}^d$. SAEs aim to unpack these into an overcomplete basis $\mathbb{R}^M$ ($M \gg d$), enabling one-to-one mapping between latent dimensions and semantic concepts.

**Encoding via Top-K Sparsity.** We utilize a Top-K SAE architecture (Gao et al., 2025). The encoder first centers the input activation $h$ and projects it into the latent space. To strictly control the sparsity density, we retain only the $k$ largest activations:

$$z = \text{TopK}\left(W_{\text{enc}}(h - b_{\text{pre}})\right). \tag{6}$$

Here, the $\text{TopK}(\cdot)$ operator acts as a dynamic thresholding mechanism, ensuring that $\|z\|_0 = k$. This contrasts with ReLU-based SAEs, where sparsity fluctuates based on activation magnitude.

**Decoding and Reconstruction.** The decoder reconstructs the input by summing the feature directions corresponding to the active latents. Let $d_j$ be the $j$-th column of the decoder matrix $W_{\text{dec}}$, representing the "feature direction" of the $j$-th semantic concept in the residual stream. The reconstruction is given by:

$$\hat{h} = b_{\text{pre}} + \sum_{j \in \mathcal{I}_k} z_j d_j, \tag{7}$$

where $\mathcal{I}_k$ is the set of indices selected by the Top-K operator.

**Training Objective.** Since sparsity is explicitly enforced by the architecture rather than a penalty term, the training objective simplifies to minimizing the Mean Squared Error (MSE) of the reconstruction:

$$\mathcal{L} = \mathbb{E}_{h \sim \mathcal{D}}\left[\|h - \hat{h}\|_2^2\right]. \tag{8}$$

We employ SAEs trained with this objective to extract $z_j$ values, which serve as the scalar indicators for the presence of safety-critical concepts in our NExT-GUARD framework.

## B.2. Statistical Metrics for Feature Selection

To identify SAE features that significantly align with unsafe content, we evaluate the statistical dependency between each feature's activation pattern and the text-level safety labels.

**Primary Metric: Variance-Normalized Mean Difference.** Our core metric quantifies the discriminative power of each feature by measuring the distributional separation between safe and unsafe samples. Given that SAE features are typically sparse and non-negative (due to the ReLU activation), we require a metric that rewards consistent high activation on unsafe inputs while penalizing instability.

Formally, for the $j$-th feature, we compute the score $s_j$ as:

$$s_j = \frac{\mu_{\text{unsafe}}^{(j)} - \mu_{\text{safe}}^{(j)}}{\sigma_{\text{unsafe}}^{(j)} + \sigma_{\text{safe}}^{(j)}}, \tag{9}$$

where $\mu^{(j)}$ and $\sigma^{(j)}$ represent the mean and standard deviation of the $j$-th feature's activations within the respective subsets ($\mathcal{D}_{safe}$ or $\mathcal{D}_{unsafe}$). This specific formulation is tailored to the properties of Sparse Autoencoders for three reasons:

1. **Encouraging Suppression in Safety:** In an ideal safety-aligned feature, activations on safe texts should be near zero (sparsity). This minimizes $\mu_{\text{safe}}^{(j)}$ and $\sigma_{\text{safe}}^{(j)}$, effectively boosting the score $s_j$.

2. **Demanding Consistency in Danger:** The numerator $\mu_{\text{unsafe}}^{(j)}$ rewards features that react strongly to unsafe content. However, SAE features often exhibit "heavy-tailed" behavior (occasional extreme outliers). A simple mean difference might select a feature that fires wildly on a single token but is silent otherwise.

3. **Penalty for Instability:** The denominator sum $\sigma_{\text{unsafe}}^{(j)} + \sigma_{\text{safe}}^{(j)}$ acts as a strict penalty for variance. It filters out "noisy" or polysemantic features that may activate on unsafe texts but fluctuate unpredictably (high $\sigma_{\text{unsafe}}^{(j)}$). This ensures that selected features are not just occasional triggers, but stable indicators of the underlying concept.

**Alternative Metrics.** To ensure that our feature selection is not an artifact of a specific metric, we cross-validated our ranking using three additional statistical methods covering classification performance, linear correlation, and information theory.

**1. F1 Score.** To evaluate the feature's potential as a binary classifier, we define a threshold-based F1 score. This metric balances the precision (avoiding over-refusal) and recall (detecting unsafe content) of the feature acting as a "trigger":

$$
\begin{aligned}
\text{Precision}_j &= \frac{\sum_{\{Y,r\}\in\mathcal{D}} \mathbb{I}(\mathbf{v}_j(Y) \geq t_j) \cdot \mathbb{I}(r=1)}{\sum_{\{Y,r\}\in\mathcal{D}} \mathbb{I}(\mathbf{v}_j(Y) \geq t_j)} \\
\text{Recall}_j &= \frac{\sum_{\{Y,r\}\in\mathcal{D}} \mathbb{I}(\mathbf{v}_j(Y) \geq t_j) \cdot \mathbb{I}(r=1)}{\sum_{\{Y,r\}\in\mathcal{D}} \mathbb{I}(r=1)} \\
s_j &= \frac{2 \cdot \text{Precision}_j \cdot \text{Recall}_j}{\text{Precision}_j + \text{Recall}_j},
\end{aligned}
\tag{10}
$$

where $\mathbb{I}(\cdot)$ is the indicator function and $t_j$ is the decision threshold.

**2. Pearson Correlation Coefficient.** To assess linear dependence, we compute the Pearson correlation coefficient $\rho_j$ between the feature activation vector $\mathbf{v}_j$ and the label vector $\mathbf{r}$:

$$
\rho_j = \frac{\text{Cov}(\mathbf{v}_j, \mathbf{r})}{\sigma_{\mathbf{v}_j}\sigma_{\mathbf{r}}} = \frac{\sum_i (v_{j,i} - \bar{v}_j)(r_i - \bar{r})}{\sqrt{\sum_i (v_{j,i} - \bar{v}_j)^2}\sqrt{\sum_i (r_i - \bar{r})^2}}.
\tag{11}
$$

**3. Mutual Information (MI).** Unlike Pearson correlation which assumes linearity, Mutual Information captures arbitrary non-linear dependencies by measuring the reduction in uncertainty about the label $r$ given the feature activation $\mathbf{v}_j$:

$$
I(\mathbf{v}_j; \mathbf{r}) = \sum_{r\in\{0,1\}} \int p(v_j, r) \log\left(\frac{p(v_j, r)}{p(v_j)p(r)}\right) dv_j.
\tag{12}
$$

We estimate this integral using the $k$-nearest neighbor estimator to handle the continuous nature of SAE activations without explicit binning.

*Remarks:* Despite the theoretical differences in these metrics—ranging from SNR-based (Mean Diff), threshold-based (F1), to information-theoretic (MI)—we observed highly consistent feature rankings in our experiments. This consistency corroborates the robustness of the features selected via our primary metric.

### B.3. Data Composition and Formatting

We aggregated the training splits of existing safety-aligned datasets. Let $\mathcal{D}_{safe}$ and $\mathcal{D}_{unsafe}$ denote the sets of safe and unsafe text samples, respectively. The initial dataset is a mixture $\mathcal{D}_{raw} = \mathcal{D}_{safe} \cup \mathcal{D}_{unsafe}$, ensuring a diverse coverage of violation categories (e.g., hate speech, self-harm, illegal acts).

To process the data uniformly, we wrapped each sample using a standardized chat template:

$$
\text{Prompt} := \texttt{User:} \quad \{p\}
\tag{13}
$$
$$
\text{Response} := \texttt{User:} \quad \{p\}\texttt{\textbackslash nAssistant:} \quad \{r\}
\tag{14}
$$

where $p$ is the user query and $r$ is the model response.

We adopted this template without special control tokens (e.g., $\texttt{<|im\_start|>}$) based on two critical observations regarding standard SAE training protocols:

1. **Training Distribution:** SAEs are typically trained on randomized text chunks derived from pre-training corpora rather than structured, turn-based dialogue contexts.

2. **Feature Stability:** Special tokens often exhibit anomalously high norms (outliers) in the residual stream. Including them can destabilize the feature extraction process, as SAEs are sensitive to such activation magnitude shifts.

Therefore, using a plain-text template ensures better alignment with the SAE's native feature space.

**Inference Masking.** Consequently, during the inference phase, our method leverages this specific Prompt/Response pattern to apply a masking strategy. We explicitly isolate the valid content, considering only the tokens corresponding to the query $\{p\}$ and the response $\{r\}$ for feature analysis, while ignoring the template scaffolding.

## C. Alternative Methods for Feature Aggregation

In the main text, we introduced NExT-Guard. Here, we detail an alternative approach, **NExT-Guard\***, which constructs a token-level classifier leveraging Sparse Autoencoder (SAE) features as noisy annotators. This method obviates the need for ground-truth token-level labels by distilling text-level supervision into token-level signals.

### C.1. Preliminaries and Feature Extraction

Let $x_{i,t}^{(L)} \in \mathbb{R}^{d_{model}}$ be the activation of the residual stream at layer $L = 18$ for the $t$-th token in the $i$-th text sequence. The SAE encoder transforms this activation into a sparse feature vector $f_{i,t} \in \mathbb{R}^{d_{feature}}$:

$$f_{i,t} = \text{ReLU}(W_{enc}(x_{i,t}^{(L)} - b_{dec}) + b_{enc}) \tag{15}$$

where $W_{enc}$, $b_{enc}$, and $b_{dec}$ are the learned parameters of the SAE.

### C.2. Text-Level Feature Selection

We treat each feature dimension $j \in \{1, \ldots, d_{feature}\}$ as a potential binary classifier for the entire text. We define the aggregation function via max-pooling over the valid response tokens $\mathcal{T}_{resp}$:

$$p_j(i) = \mathbb{I}\left(\max_{t \in \mathcal{T}_{resp}} \{f_{i,t,j}\} > 0\right) \tag{16}$$

where $\mathbb{I}(\cdot)$ is the indicator function. $p_j(i) = 1$ implies feature $j$ activated at least once in the response.

We evaluate the statistical alignment between $p_j(\cdot)$ and the ground-truth text labels $l(\cdot)$ using the **F1 score**. Based on this metric, we select two distinct feature sets:

- **Labeling Features ($\mathcal{S}_{label}$):** The top $n = 3$ features. These exhibit high precision and are treated as strong indicators of toxicity.

- **Candidate Pool ($\mathcal{S}_{pool}$):** The top $k = 10,000$ features. These serve as the rich information source for the classifier.

### C.3. Pseudo-Label Generation via Denoising

Since we lack token-level annotations, we construct pseudo-labels $y_{i,t}^{pseudo}$ by using the text-level label $l(i)$ to "denoise" the Labeling Features. A token is marked as unsafe only if a Labeling Feature activates *and* the global text context is unsafe:

$$y_{i,t}^{pseudo} = \begin{cases} 1 & \text{if } l(i) = \texttt{unsafe} \wedge \exists j \in \mathcal{S}_{label} \text{ s.t. } f_{i,t,j} > 0 \\ 0 & \text{otherwise} \end{cases} \tag{17}$$

This rule implies that while $\mathcal{S}_{label}$ features are noisy (they may activate on safe texts), their activation within a confirmed unsafe text strongly correlates with the specific harmful tokens.

### C.4. Token-Level Classifier Training

We employ a **Random Forest** classifier $\mathcal{H}$, trained to map the dense feature information from the Candidate Pool to the pseudo-labels:

$$\mathcal{H} : \{f_{i,t,j} \mid j \in \mathcal{S}_{pool}\} \to y_{i,t}^{pseudo} \tag{18}$$

We selected Random Forest for two reasons: (1) As a rank-based algorithm, it naturally handles the varying scales of SAE activations without complex normalization; (2) It captures non-linear interactions between features efficiently ($< 1\%$ compute of a linear layer backprop).

### C.5. Inference Mechanism

At inference time, the system operates as a stream guard:

1. Intercept the residual stream $x_t^{(L)}$ at layer $L$.

2. Compute SAE features $f_t$.

3. Extract the subset corresponding to $\mathcal{S}_{pool}$.

4. Feed into $\mathcal{H}$ to obtain the instantaneous safety probability.

### C.6. Performance Analysis and Future Directions

As demonstrated in the main text (see Table 1), NExT-Guard* achieves performance comparable to the primary NExT-Guard method. While this validates the feasibility of our supervised approach, the absence of significant superiority suggests that the current iteration has not yet fully capitalized on the granular, token-level information inherent in the SAE features.

However, we posit that the classifier-based paradigm remains a promising avenue. We attribute the performance plateau to the **preliminary nature** of our current classifier design, which treats each token in isolation. Future iterations could unlock greater potential by incorporating **contextual windowing**—aggregating information from preceding and succeeding tokens. Such techniques would serve to smooth the predictive signals and enforce temporal consistency, potentially mitigating the noise inherent in single-token predictions.

