# OpenReview forum: "NExT-Guard: Training-Free Streaming Safeguard without Token-Level Labels"
_ICML.cc/2026/Conference — ICML 2026 regular_

### Official Review · Reviewer_ijjS · 2026-02-17

**Soundness:** 2
**Presentation:** 4
**Significance:** 3
**Originality:** 3
**Overall Recommendation:** 4
**Confidence:** 3

**Summary:**

The authors tackle the problem of streaming safeguards for the detection of LLM misuse / harmful outputs, which focuses on classifying partial outputs as harmful/harmless, stopping harmful content before it ever reaches the end-user. Whilst existing techniques rely on token-level supervision, the authors employ SAEs to provide token-level safety signals without requiring token-level supervision at any point. The authors further show SOTA performance on a variety of standard prompt/response classification benchmarks in the streaming setting.

**Compliance With Llm Reviewing Policy:**

Affirmed.

**Final Justification:**

The authors were explicit in the ways they will revise the paper to change the claims about the need for training, and I believe the revised claims are now well supported by the experimental results.

**Key Questions For Authors:**

## [q1] Vulnerability of streaming models
Does the token-level classification approach taken by the authors create a new vulnerability that allows one to spread harms across multiple benign-looking tokens? Or instead, lead to many false positives? For example, I can imagine specific prompts in which its harmless nature is only apparent later on, with added context. Take the example of `Here is how you [kill] time waiting for a train, firstly...`. Might it be the case that streaming models treat the kill token as a false positive and block the response?

It would be insightful to see a few manually constructed examples where intent only appears later of how the proposed method handles these kinds of prompts, where safety is context-dependent; do the streaming models falsely reject them?


## [q2] WildGuardMix
The authors use the WildGuard model for evaluation, but do not perform experiments on the popular WildGuardMix dataset. Might I ask what reason there is for leaving this popular dataset out?

**Limitations:**

The limitations are addressed.

**Strengths And Weaknesses:**

# Strengths

## [s1] Timely and important problem

The authors tackle a vitally important problem. Not only would guardrail models always benefit from speed, but being able to process generated tokens iteratively before the harmful response is seen by the user is a genuinely useful additional property. The authors' contribution to this line of research is therefore of high value to both researchers and practitioners alike.

## [s2] Exhaustive evaluation, and strong performance

The experimental comparisons made by the authors are thorough. The authors evaluate the performance across a wide range of datasets and guardrail models; both of the full model response and streaming variety. The authors' two method variants are clearly the SOTA in the streaming regime.


# Weaknesses

## [w1] Questionable claims; the method is not training-free

The authors highlight that a key advantage of their method over prior work is that is is "training-free" (in the abstract [L024], introduction [L106, L117], "in a completely training-free manner." [L096]). Furthermore, [L090] states that "NEXT-GUARD does not require training SAEs from scratch".

I believe that at least the first claim (and possibly the second) are false, and could be misleading.

1. Firstly, on [L133], the authors detail the necessary training of the contrastive step of their method on publicly available, labeled datasets. This is not training free in any reasonable definition of the term, and actually requires a corpus of labeled samples of harmful and harmless requests/responses. However, it is true that the method does not require token-level supervision--which may indeed be a strength--but this is a very different property than being "completely training-free".
2. Furthermore, in the experiments on [L253], the authors appear to state that the proposed Next-guard requires an SAE to be trained from scratch on the guard models. `We construct our NEXTGUARD by training Sparse Autoencoder on Qwen3Guard8B-Gen`  Might the authors expand on whether this was a typo, and point to the public resources used in this case, on which the authors' argument rests? I am not aware of any existing SAEs trained on such GuardModels. If SAEs need to be trained on guard models, the authors' pipeline inherits an extra burden; how we do ensure SAEs are trained sufficiently well to capture the relevant safety features in the first place? What training data do we use?, etc.

Ultimately, this raises the question about the true "cost" of adopting the proposed methodology, and the assumptions about necessary data/supervision, which I worry are overly exaggerated in the introduction and abstract, in a way that does not reflect the experiments.


## [w2] Important missing experimental dataset/evaluation details

The authors state they use a "random" subset of data D to train the contrastive part of their method [L133]. However, there is no information I can find about this procedure -- Appendix B.3 expands on how these datasets are combined, but gives no information about which specific datasets are used, for which experiments. The authors also state [L137] that D is "robust" to the specific choice of D, but I do not find experiments confirming this.

My concern here is that strong results may depend a lot on how these datasets are chosen/combined--these details should be explained fully. For example, do the authors use samples from all datasets (on which they later evaluate in table 1), as sample-level supervision? It would be useful to see the results on the streaming classification of particular datasets when leaving entire specific dataset subsets out from the training data, to ensure this generalizes, and is not overfitting (the authors reasonably state overfitting is a concern in streaming guardrail models). Furthermore, can the authors clarify that the results in Table 1 are on a held-out test set, given that samples from these datasets are used to learn the safety-relevant SAE features at train-time?


**Summary**: whilst i have a negative initial review based on the above, I would be open to increasing my score if these issues and concerns were adequately addressed.


## minor:
* [L124] -> "Additionally", "we present"

---

> ### Author Rebuttal · Authors · 2026-03-31
>
> Dear Reviewer ijjS,
>
> Thank you for your constructive feedback! We are greatly encouraged by your recognition that our work `tackles a vitally important problem` and that our contribution `is of high value to both researchers and practitioners`. Below, we provide our point-by-point responses. Hope our response could address your concerns!
>
> ---
> ### **1. Clarification on the Term ``Training-free’’**
> (*``W1. Questionable Claims: …the method does not require token-level supervision, but this is different than being completely training-free.’’*)
>
> Thank you for this insightful correction. We agree that our initial use of ``training-free’’ was imprecise. While our method is the first to achieve streaming safety without *token-level supervision* and *gradient-based fine-tuning*, we recognize that the underlying SAE still involves a training process.
>
> Following your feedback, we have thoroughly revised the manuscript to ensure technical accuracy. For instance,
> - Title: `NEXT-Guard: Training-Free Streaming Safeguard...` →`NEXT-Guard: Label-Free Streaming Safeguard...`
> - Abstract [L24]: `...a training-free framework that...` → `...a label-free framework that...`
> - Introduction [L96]: `...in a completely training-free manner.` → `...without the requirement for token-level labels.`
> - Method [L107/117]: Removed `training-free` and replaced it with the `label-free` or `plug-and-play`.
>
> By refining our terminology, we believe the paper now more accurately communicates its core contribution: eliminating the prohibitive cost of token-level annotations.
>
> Hope these corrections could address your concerns!
>
> ---
> ### **2. Evaluation Details and Dataset Generalizability**
> (*W2. Missing Evaluation Details: ``The authors use dataset $D$ to train the contrastive part…details should be explained fully... to ensure this generalizes.’’*)
>
> Thank you for your suggestions!
> 1. The dataset $D$ consists of 300 random samples drawn from Aegis2.0 training set, and is fixed across all experiments.
> 2. While $D$ is drawn from Aegis2.0, NExT-Guard achieves SOTA performance on all other datasets.
> 3. To further address your concern, we added experiments by replacing Aegis2.0 with BeaverT and Aegis:
> ||||||||||
> |-|:-:|:-:|:-:|:-:|:-:|:-:|:-:|:-:|
> |||Prompt||||Response|||
> ||Aegis|Aegis2.0|SimpST|**Avg.**|SafeRLHF|BeaverT|Aegis2.0|**Avg.**|
> |SCM|73.8|78.6|96.5|**83.0**|59.8|83.4|80.4|**74.5**|
> |Kelp|73.5|78.2|96.7|**82.8**|54.0|84.1|81.5|**73.2**|
> |NExT-Guard (Aegis2.0)|88.5|84.8|98.0|**90.4**|83.3|80.6|82.0|**82.0**|
> |NExT-Guard (BeaverT)|87.8|84.2|98.1|**90.0**|83.0|80.0|81.3|**81.4**|
> |NExT-Guard (Aegis)|88.8|81.9|93.8|**88.1**|80.5|78.1|79.0|**79.2**|
> ||||||||||
>
> The results demonstrate the **robustness** of NExT-Guard, and we have incorporated all the above additions into the new version.
>
> Hope our response could address your concerns!
>
> ---
> ### **3. Vulnerability of Streaming Models**
> (*Q1: ``Does it create a new vulnerability that spreads harms across multiple benign tokens? Or instead, lead to false positives?…’’*)
>
> Thank you for this insightful question!
>
> The vulnerability you mentioned is indeed a **core issue of existing token-level supervised** streaming safeguards. Such methods tend to overfit to specific tokens (e.g., Kill), leading to both false positives and false negatives. For example (real cases with Qwen3-Guard-Stream):
> - *Here is how you kill time waiting for a train* → falsely flagged ``kill'' as unsafe
>
> - *How to silence my noisy boss forever* → falsely flagged as safe
>
> These errors arise because token-level supervision encourages the model to treat certain words as inherently unsafe/safe, ignoring context.
>
> **How NExT-Guard Addresses This:**
>
> It avoids this issue by relying on semantic latent features rather than surface tokens:
>
> - *…kill time waiting for a train…* → correctly safe
>
> - *I want to kill my boss.* → correctly flagged ``boss'' as unsafe
>
> - *… silence my noisy boss forever* → correctly flagged ``forever'' as unsafe
>
> This further supports our key claim: moving beyond token-level supervision is crucial not only for efficiency, but also for robustness in real streaming safety.
>
> Hope this response could meet your expectations!
>
> ---
> ### **4. Reason for Leaving WildGuardMix Dataset  Out**
> (*Q2: ``Might I ask what reason for leaving WildGuardMix out?’’*)
>
> The reason we initially did not include WildGuardMix is that we believe our datasets are already standard in streaming safety and provide sufficient diversity and coverage.
>
> To address your concern, we have added WildGuardMix in the new version:
>
> ||||||
> |-:|-:|-:|-:|-:|
> |LlamaGuard3-8B: 76.4| |Qwen3Guard-8B: 84.7| |SCM: 72.5|
> |LlamaGuard4-12B: 73.0| |Qwen3Guard-4B: 85.3| |Kelp: 83.9|
> |WildGuard: 88.9| |Qwen3Guard-0.6B: 86.0| |NExT: **86.3**|
> ||||||
>
> NExT-Guard outperforms all other benchmarks, second only to WildGuard, demonstrating its generalization ability.
>
> ---
> Once again, we are deeply appreciative of the time and expertise you have shared with us!
>
> Best,
>
> Authors

---

> > ### Author Rebuttal · Reviewer_ijjS · 2026-04-01
> >
> > Thanks to the authors for the effort with the rebuttal!
> >
> > I appreciate the authors making explicit the ways in which they change the paper's claims around the terminology, and now i now think the results now support the paper's (revised) claims. I will increase my score accordingly.

---

> > > ### Author Response · Authors · 2026-04-02
> > >
> > > Dear Reviewer ijjS,
> > >
> > >
> > > Thank you very much for your positive feedback! We are truly encouraged by your recognition of our work. As you suggested, we will ensure that all the details are explicitly documented in the revised manuscript.
> > >
> > > We would like to take this opportunity to reiterate the core contribution of our paper:
> > > - ### *While current academic and industrial paradigms primarily rely on token-level supervision, we prove that such dense label is NOT indispensable for achieving effective streaming safety and may even introduce overfitting risks.*
> > >
> > > We believe this insight offers an insightful and critical perspective for the burgeoning field of streaming safety, providing a more robust and label-free alternative for streaming protection.
> > >
> > > Thank you again for your thoughtful comments throughout the review process. We are fully committed to advancing the field of streaming safety and contributing to the community. Your feedback and support are invaluable to us in achieving this goal.
> > >
> > > Best regards,
> > >
> > > Authors of paper 32031

---

### Official Review · Reviewer_WfX9 · 2026-03-07

**Soundness:** 2
**Presentation:** 2
**Significance:** 3
**Originality:** 2
**Overall Recommendation:** 4
**Confidence:** 4

**Summary:**

The paper proposes NExT-GUARD, a training-free framework designed for streaming LLM safeguards. Unlike existing streaming guards that require expensive token-level supervised training and are prone to keyword-based overfitting, the authors argue that safety signals are already latent within the hidden representations of post-hoc safeguards. By using Sparse Autoencoders (SAEs), NExT-GUARD identifies these safety-critical features offline through a contrastive analysis and monitors their activations in real-time during generation. The authors provide a theoretical justification for early intervention using the Martingale Optional Stopping Theorem and demonstrate that their method achieves state-of-the-art F1 scores across multiple benchmarks.

**Compliance With Llm Reviewing Policy:**

Affirmed.

**Final Justification:**

After reading the rebuttal, I appreciate the additional experiments (e.g., FPR/FSR, Llama3 results, and ablations), which strengthen the empirical evaluation. While some core concerns remain only partially resolved (particularly the monotonicity assumption), I have increased my confidence in the paper’s empirical contribution.

**Key Questions For Authors:**

1. **Contextual Reversal:** How does the "Temporal Semantic Monotonicity" assumption handle cases where an initially "unsafe" token is rendered safe by the end of the sequence (e.g., "I am killing... time")?
2. **False Positive Analysis:** Why was the False Positive Rate (FPR) omitted? Can you provide the over-refusal rate for benign datasets?
3. **Dataset Balance:** What was the specific safe-to-unsafe ratio in the calibration and test sets? Is the high F1-score robust to real-world class imbalance?
4. **Ablation on K:** Can you provide a sensitivity analysis for the choice of $K=32$ features? Does increasing $K$ lead to a prohibitive increase in FPs?

**Limitations:**

* **Scope:** The evaluation primarily focuses on the Qwen series as the base model to ensure controlled comparisons, leaving questions about generalizability across broader model architectures.
* **Linguistic Nuance:** The system may be prone to over-refusal in cases of non-literal or metaphorical language due to the reliance on the monotonicity assumption.

**Strengths And Weaknesses:**

### Strengths:
* **Efficiency:** The "training-free" nature allows for low-cost deployment by leveraging publicly available SAEs, bypassing the need for gradient updates or labor-intensive token-level annotations.
* **Performance:** Empirical results show high F1-scores (90.8 on prompts, 84.3 on responses), reportedly outperforming industrial-grade supervised streaming baselines like Qwen3Guard-Stream.
* **Interpretability:** By mapping risks to specific SAE features (e.g., feature #4744 for criminal content), the model offers mechanistic transparency into intervention decisions.

### Weaknesses
* **Flawed Theoretical Assumption (Soundness):** The mathematical proof for intervention sufficiency relies on "Temporal Semantic Monotonicity," assuming harmful intent is only ever amplified and never diluted. This fails to account for "contextual reversal" (e.g., "killing time"), where early unsafe tokens are neutralized by subsequent context.
* **Incomplete Evaluation Metrics:** The paper relies on F1-scores but lacks a dedicated **False Positive Rate (FPR)** or Over-refusal table. In streaming scenarios, the risk of prematurely stopping benign sentences is a critical bottleneck that is not quantitatively addressed.
* **Novelty:** While the specific "training-free" framing is distinct, the use of SAEs for safety steering and monitoring is explored in related literature [1, 2], making the "first-of-its-kind" claim potentially overstated. Similar approaches like Sparse Representation Steering (SRS) already utilize pre-trained SAEs to identify and act on safety-relevant directions.
* **Hidden Data Dependency:** Although labeled "training-free," the method requires an offline "calibration phase" using labeled samples to rank features. The system's objectivity is thus strictly limited by the quality of this initial data.
* **Presentation Inconsistencies:** Several visualizations lack clarity. Figure 3 contains inconsistent numerical logic (e.g., showing 0.7/0.5 bars but 0.6 later), and the "grey" SAE activation plots in Figure 6 are difficult to interpret.

[1] He, J., et al. "Interpretable LLM Guardrails via Sparse Representation Steering." arXiv:2503.16851, 2025.

[2] Yeon, J., et al. "GSAE: Graph-regularized sparse autoencoders for robust llm safety steering." arXiv:2512.06655, 2025.

---

> ### Author Rebuttal · Authors · 2026-03-31
>
> Dear Reviewer WfX9,
>
> Thank you for your constructive feedback! Below, we have summarized your concerns into seven points. Hope our response could address your concerns!
>
> ---
>
> ### **1. Add More Base LLMs**
> (*L1: ``The evaluation primarily focuses on Qwen, leaving questions about generalizability across broader models’’*)
>
> Thank you for your concern! Following your comments, we have added Llama3-8B in the new version:
>
> ||||||||||
> |-|:-:|:-:|:-:|:-:|:-:|:-:|:-:|:-:|
> |||Prompt||||Response|||
> ||Aegis|Aegis2.0|SimpST|Avg.|SafeRLHF|BeaverT|Aegis2.0|Avg.|
> |SCM|73.8|78.6|96.5|83.0|59.8|83.4|80.4|74.5|
> |Kelp|73.5|78.2|96.7|82.8|54.0|84.1|81.5|73.2|
> |NExT (Llama3)|85.1|79.6|96.9|**87.2**|79.0|82.7|79.2|**79.6**|
> |NExT (Qwen3)|88.5|84.8|98.0|**90.4**|83.3|80.6|82.0|**82.0**|
> ||||||||||
>
> The results show that NExT-Guard still achieves competitive performance, demonstrating the generalizability of our paradigm.
>
> ---
> ### **2. Add Experiments about False Positive Rate**
> (*W2 & Q2: ``The paper relies on F1-scores but lacks a dedicated FSR…’’*)
>
> For fairness, our initial evaluation follows the current streaming safety works (e.g., SCM, Kelp), which focus on F1 scores.
>
> To address your concern, we have added FSR in the new version:
> ||||||
> |-|:-:|:-:|:-:|:-:|
> ||Aegis|Aegis2.0|SimpST|**Avg.**|
> |SCM|0.22|0.20|0.10|0.17|
> |Kelp|0.24|0.25|0.11|0.20|
> |NExT-Guard|0.09|0.10|0.04|**0.07**|
> ||||||
>
> ---
> ### **3. Handling Non-Monotonic Context**
> (*L2 & W1 & Q1: ``How to handle cases where an initially unsafe text is rendered safe by the end of the sequence?’’*)
>
> Thank you for your concern. While a sequence might transition from Unsafe to Safe, the initial exposure of harmful content already constitutes a safety breach in a streaming context. For example, a classic jailbreak text is:
> - Describing how to make a bomb then condemning it.
>
> From a risk-mitigation perspective, we believe it is safer and more reasonable to maintain an unsafe classification.
>
> ---
> ### **4. The Difference from Current Works**
> (*W3: `` The use of SAEs for safety steering is explored in related literature [1, 2]...’’*)
>
> We respectfully clarify that NExT-Guard is fundamentally different from [1-2] you mentioned. Specifically,
> ||||
> |-|-|-|
> |Aspect|[1-2]|NExT-Guard|
> |Objective|Steering|Real-time Safeguard|
> |Core Insight|SAE features can steer LLMs|Token-level supervision is NOT essential for streaming safety|
> |Experiments|Steer *vs.* SFT/RL|Token-level supervision *vs.* SAE|
> |Theoretical Focus|Representation Interpretability|Probabilistic lower-bound for streaming safety|
> |Technical Focus|How to steer outputs|Extract signals for streaming (NOT post-hoc) safety |
> ||||
>
> To address your concern, in the new version, we have analyzed the importance of [1-2] to the community while highlighting their technical differences from our work.
>
> ---
> ### **5. Add Experiments on Hyperparameter**
> (*Q4: ``Can you provide a sensitivity analysis for K?’’*)
>
> Sure! We have added the hyperparameter experiments:
> ||||||
> |-|:-:|:-:|:-:|:-:|
> ||Aegis|Aegis2.0|SimpST|**Avg.**|
> |Kelp|73.5|78.2|96.7|82.8|
> |K=24|78.4|74.2|81.6|**78.1**|
> |K=28|83.1|79.5|88.4|**83.7**|
> |K=32|88.5|84.8|95.0|**89.4**|
> |K=36|82.3|84.1|90.2|**86.5**|
> |K=40|83.6|79.4|88.3|**83.8**|
> ||||||
>
> ||||||
> |-|:-:|:-:|:-:|:-:|
> ||SafeRLHF|BeaverT|Aegis (R)|**Avg.**|
> |Kelp|54.0|84.1|81.5|73.2|
> |K=24|72.4|69.8|75.3|**72.5**|
> |K=28|78.1|76.3|81.4|**78.6**|
> |K=32|83.3|80.6|82.0|**82.0**|
> |K=36|84.5|80.2|79.1|**81.3**|
> |K=40|79.4|77.0|80.3|**78.9**|
> ||||||
>
> As show, our choice of K = 32 is optimal in most cases.
>
> ---
> ### **6. Dependency on Calibration Dataset D**
> (*W4 & Q3: ``The system's objectivity is limited by the quality of $D$’’*)
>
> In practical, NExT-Guard is highly robust to the construction of the $D$. To prove this, in the new version, we have added experiments which replace it with BeaverT and Aegis:
>
> ||||||||||
> |-|:-:|:-:|:-:|:-:|:-:|:-:|:-:|:-:|
> |||Prompt||||Response|||
> ||Aegis|Aegis2.0|SimpST|Avg.|SafeRLHF|BeaverT|Aegis2.0|Avg.|
> |Kelp|73.5|78.2|96.7|82.8|54.0|84.1|81.5|73.2|
> |NExT-Guard (Aegis2.0)|88.5|84.8|98.0|**90.4**|83.3|80.6|82.0|**82.0**|
> |NExT-Guard (BeaverT)|87.8|84.2|98.1|**90.0**|83.0|80.0|81.3|**81.4**|
> |NExT-Guard (Aegis)|88.8|81.9|93.8|**88.1**|80.5|78.1|79.0|**79.2**|
> ||||||||||
>
> Additionally, the safe-to-unsafe ratio in both $D$ and our test datasets remains consistent with experiments in previous works, ensuring a fair comparison.
>
> ---
>
> ### **7. Visualization Clarity**
> (*W5: ``Several visualizations lack clarity (i.e., 0.6 in Figure 3 and activation plots in Figure 6)’’*)
>
> We have revised the original version as follows:
> - Figure 3: Adding a caption note to explicitly clarify that 0.6 is derived as a average of 0.7 and 0.5.
> - Figure 6: Further highlighting the semantics of activation plot that were originally provided in the side legend.
>
> Thank you for helping us improve the quality of our work!
>
> ---
>
> Once again, we are deeply appreciative of the time and expertise you have shared with us!
>
> Best,
>
> Authors

---

> > ### Author Rebuttal · Reviewer_WfX9 · 2026-04-03
> >
> > I thank the authors for the detailed rebuttal and the extensive additional experiments. The inclusion of the False Positive Rate (FPR) evaluation, the Llama3-8B results, and the ablations significantly strengthen the empirical standing and generalizability of the paper.
> >
> > Regarding my theoretical concern about the "Temporal Semantic Monotonicity" assumption, I appreciate the practical risk-mitigation argument that initial exposure of harmful content constitutes a breach in a strict streaming context. This adequately resolves the issue for the scope of this work. However, because this assumption naturally limits the framework's ability to handle contextual reversals (e.g., non-literal language or benign neutralizing context), I ask that this structural limitation be explicitly discussed in the limitations section of the camera-ready version.
> >
> > Overall, the strong rebuttal and additional empirical evidence have fully addressed my concerns. I will be maintaining my positive score and increasing my confidence rating to reflect the improved evaluation.

---

> > > ### Author Response · Authors · 2026-04-04
> > >
> > > Dear Reviewer WfX9,
> > >
> > >
> > >
> > > Thank you very much for your positive feedback! We are truly encouraged by your recognition of our work. As you suggested, we will ensure that all the details are explicitly documented in the revised manuscript.
> > >
> > > Regarding the ``temporal semantic monotonicity'' assumption:
> > > - We sincerely appreciate your insightful perspective on contextual reversals! As you suggested, we will explicitly document these additions in the `Discussion` section of our camera-ready version. We agree that incorporating this discussion will provide a more balanced and comprehensive view of the NExT-Guard framework.
> > >
> > > Furthermore, we would like to take this opportunity to reiterate the core contribution of our paper:
> > > - **While current academic and industrial paradigms primarily rely on token-level supervision, we prove that such dense label is NOT indispensable for achieving effective streaming safety and may even introduce overfitting risks.**
> > >
> > > We believe this insight offers an insightful and critical perspective for the burgeoning field of streaming safety, providing a more robust and label-free alternative for streaming protection.
> > >
> > > Thank you again for your thoughtful comments throughout the review process. We are fully committed to advancing the field of streaming safety and contributing to the community. Your feedback and support are invaluable to us in achieving this goal.
> > >
> > > Best regards,
> > >
> > > Authors of paper 32031

---

### Official Review · Reviewer_Fyx7 · 2026-03-09

**Soundness:** 3
**Presentation:** 3
**Significance:** 3
**Originality:** 3
**Overall Recommendation:** 4
**Confidence:** 3

**Summary:**

The paper introduces NExT-Guard, a training-free streaming safeguard designed to intercept unsafe large language model (LLM) generations in real time. Conventional post-hoc safeguards evaluate safety only after a full sequence is generated, which can introduce a temporal delay before harmful content is detected. Meanwhile, existing streaming safeguards often rely on token-level supervised training, which requires costly annotations and may risk overfitting.

To address this, NExT-Guard leverages pre-trained Sparse Autoencoders (SAEs) to disentangle the hidden states of existing post-hoc models into sparse and interpretable safety-related features. By identifying risk-relevant features offline and aggregating their activations, the system computes a real-time risk score that can be used to intervene during text generation.

**Compliance With Llm Reviewing Policy:**

Affirmed.

**Final Justification:**

After rebuttal, my concerns are justified.

**Key Questions For Authors:**

1. **Calibration Dataset Details:** Please clarify which datasets and splits were used to construct the calibration dataset \(D\). Can the authors confirm that there is no overlap with the evaluation test sets used in Table 1?

2. **Threshold Selection:** How were the predefined thresholds for \(c_t\) and the selection of \(K = 32\) determined? Were these values fixed across all experiments or tuned separately per dataset?

3. **Handling Non-Monotonic Context:** How does NExT-Guard behave in cases where an early safety-feature activation is later neutralized by benign context? More generally, how robust is the method to abrupt semantic changes during generation (e.g., negation or contextual clarification), which may violate the monotonic or bounded-drift assumptions used in the theoretical analysis?

4. **Annotation Protocol:** Could the authors provide more details about the annotation process used to identify the first unsafe token in the 300 sampled responses? Specifically, information about annotator background, annotation guidelines, and inter-annotator agreement would help assess the reliability of the constructed token-level ground truth.

5. **Model Generalization:** The evaluation primarily focuses on the Qwen model family. Have the authors evaluated NExT-Guard on other architectures (e.g., Llama or Mistral), or do they expect the approach to generalize similarly across different model families?

**Limitations:**

yes

**Strengths And Weaknesses:**

**Strengths**

* **Alternative Safeguarding Paradigm:** The work explores an alternative direction that avoids training new token-level classifiers and instead attempts to extract safety signals from representations already present in frozen post-hoc models. This approach may offer practical advantages by reducing the need for additional training.

* **Strong Reported Performance:** The authors report competitive detection performance across multiple benchmarks, achieving an average F1 of 90.8 on prompt classification and 84.3 on response classification, outperforming several streaming and post-hoc baselines in their experiments.

* **Intervention Timing:** Compared with token-supervised baselines that may trigger early due to keyword correlations, the reported intervention distribution for NExT-Guard appears to more closely align with the annotated onset of unsafe tokens.

**Weaknesses & Methodological Concerns**

While the core idea is interesting, several aspects of the experimental design and theoretical assumptions remain unclear, which makes it difficult to fully assess the reliability of the reported results.

* **Unclear Calibration Data Construction:** In Stage 1, the authors construct a calibration dataset \(D\) by "randomly extracting safe and unsafe samples from publicly available safety benchmarks." However, the paper does not specify which benchmarks were used. If samples from the evaluation datasets (Aegis, Aegis2.0, SafeRLHF, BeaverTails) were used to select the safety features, this could introduce potential data leakage that may affect the reported results.

* **Limited Transparency in Hyperparameter Selection:** NExT-Guard triggers intervention when a risk score \(c_t\) exceeds a predefined threshold and also relies on selecting the top \(K\) features (typically 32). The manuscript does not clarify whether these hyperparameters were fixed globally or tuned separately for each dataset. If tuned per dataset, the comparisons with baselines may not be fully controlled.

* **Human Annotation Protocol:** To evaluate early intervention, the authors construct token-level ground truth by asking human annotators to identify the first unsafe token in 300 responses. However, the paper does not provide details about the annotators, annotation guidelines, or agreement metrics. Additional transparency regarding the annotation process would improve confidence in these labels.

* **Limited Model Diversity:** Although the framework is presented as broadly applicable, the evaluation primarily focuses on the Qwen model family (Qwen3Guard-8B and Qwen3-8B). Experiments on additional architectures such as Llama or Mistral would help support claims of broader generality.

* **Assumption of Temporal Semantic Monotonicity:** In Appendix D.2.1, the proof for Intervention Sufficiency assumes that safety-related features behave as a sub-martingale, implying that unsafe intent tends to be preserved or amplified during generation. However, language generation is often non-monotonic. For instance, an apparently harmful phrase (e.g., "To execute the kill") may later be clarified in a benign programming context ("command in Python").

* **Bounded Semantic Drift Assumption:** The application of Azuma-Hoeffding's inequality relies on the assumption of bounded semantic drift between consecutive tokens. In practice, natural language can exhibit abrupt semantic changes (for example, through negation or contextual clarification), which may challenge the validity of this assumption.


This paper presents an interesting approach to real-time LLM safety by attempting to extract safety signals from sparse representations within existing models. The idea of leveraging SAE features for streaming safeguards is potentially promising and may offer practical advantages in settings where additional training is undesirable.

However, the current manuscript leaves several important methodological details unclear. In particular, the construction of the calibration dataset raises potential concerns about dataset overlap, and the lack of transparency regarding hyperparameter selection makes it difficult to fully evaluate the fairness of the reported comparisons. In addition, the theoretical analysis relies on assumptions that may not fully reflect the non-monotonic behavior of natural language generation.

Clarifying the experimental setup, providing additional details on the calibration process and annotation protocol, and discussing the limitations of the theoretical assumptions would significantly strengthen the paper. I would be open to revisiting my evaluation if these concerns are addressed during rebuttal.

---

> ### Author Rebuttal · Authors · 2026-03-30
>
> Dear Reviewer Fyx7,
>
> Thank you for your constructive feedback! Below, we have summarized your concerns into four points. Hope our response could address your concerns!
>
> ---
>
> ### **1. How to Choose Calibration Dataset?**
> (*W1 & Q1: ``Please clarify which datasets were used to construct calibration dataset... Can the authors confirm there is no overlap with the test sets?’’*)
>
> Thank you for your concern. We can confirm that there is **NO overlap** with the test sets, as the calibration dataset was constructed using the train set in Aegis2.0, which was not used in the evaluations.
>
> To further address your concern, in the new version, we have added experiments which replace it with BeaverT and Aegis:
>
> ||||||||||
> |-|:-:|:-:|:-:|:-:|:-:|:-:|:-:|:-:|
> |||Prompt||||Response|||
> ||Aegis|Aegis2.0|SimpST|**Avg.**|SafeRLHF|BeaverT|Aegis2.0|**Avg.**|
> |SCM|73.8|78.6|96.5|**83.0**|59.8|83.4|80.4|**74.5**|
> |Kelp|73.5|78.2|96.7|**82.8**|54.0|84.1|81.5|**73.2**|
> |NExT-Guard (Aegis2.0)|88.5|84.8|98.0|**90.4**|83.3|80.6|82.0|**82.0**|
> |NExT-Guard (BeaverT)|87.8|84.2|98.1|**90.0**|83.0|80.0|81.3|**81.4**|
> |NExT-Guard (Aegis)|88.8|81.9|93.8|**88.1**|80.5|78.1|79.0|**79.2**|
> ||||||||||
>
> The results show that our method is highly robust to the choice of calibration dataset.
>
> ### **2. Additional Hyperparameter Experiments & Details of Annotation Process.**
> (*W2 & W3 & Q2 & Q4: ``How were the predefined c and K determined?… For annotation process, more details such as annotation guidelines would improve confidence.’’*)
>
> The values of c = 0.8 and K = 32 were determined through hyperparameter experiments and are fixed globally. To address your concern, we have added the hyperparameter experiments:
>
> ||||||
> |-|:-:|:-:|:-:|:-:|
> ||Aegis|Aegis2.0|SimpST|**Avg.**|
> |SCM|73.8|78.6|96.5|83.0|
> |Kelp|73.5|78.2|96.7|82.8|
> |c=0.5|68.4|69.2|75.1|**70.9**|
> |c=0.6|72.1|76.3|78.9|**75.8**|
> |c=0.7|80.2|78.6|89.4|**82.7**|
> |c=0.8|88.5|84.8|95.0|**89.4**|
> |c=0.9|74.6|85.3|85.5|**81.8**|
> ||||||
>
> ||||||
> |-|:-:|:-:|:-:|:-:|
> ||SafeRLHF|BeaverT|Aegis (R)|**Avg.**|
> |SCM|59.8|83.4|80.4|**74.5**|
> |Kelp|54.0|84.1|81.5|**73.2**|
> |c=0.5|53.1|70.6|75.1|**66.2**|
> |c=0.6|68.5|74.8|76.2|**73.2**|
> |c=0.7|73.1|76.2|79.4|**78.5**|
> |c=0.8|83.3|80.6|82.0|**82.0**|
> |c=0.9|79.4|77.0|80.3|**79.8**|
> ||||||
>
> Additionally, following your suggestions, K's hyperparameter experiments have been added:
>
> ||||||
> |-|:-:|:-:|:-:|:-:|
> ||Aegis|Aegis2.0|SimpST|**Avg.**|
> |SCM|73.8|78.6|96.5|83.0|
> |Kelp|73.5|78.2|96.7|82.8|
> |K=24|78.4|74.2|81.6|**78.1**|
> |K=28|83.1|79.5|88.4|**83.7**|
> |K=32|88.5|84.8|95.0|**89.4**|
> |K=36|82.3|84.1|90.2|**86.5**|
> |K=40|83.6|79.4|88.3|**83.8**|
> ||||||
>
> ||||||
> |-|:-:|:-:|:-:|:-:|
> ||SafeRLHF|BeaverT|Aegis (R)|**Avg.**|
> |SCM|59.8|83.4|80.4|74.5|
> |Kelp|54.0|84.1|81.5|73.2|
> |K=24|72.4|69.8|75.3|**72.5**|
> |K=28|78.1|76.3|81.4|**78.6**|
> |K=32|83.3|80.6|82.0|**82.0**|
> |K=36|84.5|80.2|79.1|**81.3**|
> |K=40|79.4|77.0|80.3|**78.9**|
> ||||||
>
> Regarding the annotation process, we have now provided detailed settings (e.g., annotator background & guidelines) in Appendix. For example:
> 1. Definition of Unsafe: Unsafe words include harmful, explicit, or potentially dangerous terms like violence, hate speech, and illegal activities.
> 2. Annotator's Role: Annotators should read the sentence and select the first unsafe word; If no unsafe word is found, label as ``No unsafe word found''.
>
> ### **3. Experiments on More Base LLMs.**
> (*W4 & Q5: ``Experiments on additional architectures such as Llama or Mistral would help support claims of broader generality.’’*)
>
> Thank you for your suggestion! We agree that testing on more base LLMs is valuable. Following your recommendation, we have added Llama3-8B as a base model:
>
> ||||||||||
> |-|:-:|:-:|:-:|:-:|:-:|:-:|:-:|:-:|
> |||Prompt||||Response|||
> ||Aegis|Aegis2.0|SimpST|**Avg.**|SafeRLHF|BeaverT|Aegis2.0|**Avg.**|
> |SCM|73.8|78.6|96.5|**83.0**|59.8|83.4|80.4|**74.5**|
> |Kelp|73.5|78.2|96.7|**82.8**|54.0|84.1|81.5|**73.2**|
> |NExT (Llama3)|85.1|79.6|96.9|**87.2**|79.0|82.7|79.2|**79.6**|
> |NExT (Qwen3)|88.5|84.8|98.0|**90.4**|83.3|80.6|82.0|**82.0**|
> ||||||||||
>
> The results show that NExT-Guard still achieves competitive performance, demonstrating the robustness of our paradigm.
>
> ---
>
> ### **4. Handling Non-Monotonic Context.**
> (*W5 & W6 & Q3: ``How does NExT-Guard behave where an early unsafe sequence is later neutralized by benign context?’’*)
>
> Thank you for your concern. While a sequence might transition from Unsafe to Safe, the initial exposure of harmful content already constitutes a safety breach in a streaming context. For example,
> - Describing how to make a bomb then condemning it.
>
> From a risk-mitigation perspective, we believe it is safer and more reasonable to maintain an unsafe classification.
>
> Following your feedback, we have added the above clarification in the new version. Hope our response could address your concern!
>
> ---
>
> Once again, we are deeply appreciative of the time and expertise you have shared with us!
>
> Best,
>
> Authors

---

> > ### Author Rebuttal · Reviewer_Fyx7 · 2026-04-03
> >
> > Thanks for the rebuttal.

---

> > > ### Author Response · Authors · 2026-04-03
> > >
> > > Dear Reviewer Fyx7,
> > >
> > >
> > > Thank you very much for your positive feedback! We are truly encouraged by your recognition of our work. As you suggested, we will ensure that all the details are explicitly documented in the revised manuscript.
> > >
> > > We would like to take this opportunity to reiterate the core contribution of our paper:
> > > - ### *While current academic and industrial paradigms primarily rely on token-level supervision, we prove that such dense label is NOT indispensable for achieving effective streaming safety and may even introduce overfitting risks.*
> > >
> > > We believe this insight offers an insightful and critical perspective for the burgeoning field of streaming safety, providing a more robust and label-free alternative for streaming protection.
> > >
> > > Thank you again for your thoughtful comments throughout the review process. We are fully committed to advancing the field of streaming safety and contributing to the community. Your feedback and support are invaluable to us in achieving this goal.
> > >
> > > Best regards,
> > >
> > > Authors of paper 32031

---

### Official Review · Reviewer_PeiJ · 2026-03-13

**Soundness:** 2
**Presentation:** 3
**Significance:** 2
**Originality:** 2
**Overall Recommendation:** 3
**Confidence:** 4

**Summary:**

This paper uses sparse autoencoders (SAEs) to mine feature dimensions related to unsafe content from the hidden representations of existing post-hoc safeguards, and computes a final risk score, thereby improving post-hoc protection. The experiments show that this method outperforms existing streaming safeguards and some post-hoc safeguards on multiple prompt and response safety benchmarks.

**Compliance With Llm Reviewing Policy:**

Affirmed.

**Final Justification:**

Thank you to the authors for their effort in responding, which addressed most of my concerns. However, some issues still remain. For example, the time and computational overhead of NExT-Guard during training and inference is still unclear. On the one hand, in the rebuttal, the authors only reported the training-time overhead and did not provide any comparison on inference. On the other hand, in the reply to the rebuttal comments, they only stated that NExT-Guard requires less computation overall for training, without offering substantive experimental comparisons. However, detailed clarification of the time and resource consumption of SAE-based Qwen3Guard (i.e., NExT-Guard), Qwen3Guard-Gen, and Qwen3Guard-Stream, especially at inference time, is very important for real-time safeguarding. In addition, I suggest that the authors release the final code. Overall, I have decided to raise my score from 2 to 3. Even so, I hope the AC will take the other reviewers’ opinions into full consideration as well.

**Key Questions For Authors:**

1. This paper seems to only select the dimensions in SAE features that are related to unsafe content. Do the authors know, and how do they determine, what feature meanings these unsafe dimensions represent?
2. Beyond fixed safety benchmarks, how effective is the proposed method for safeguarding real language generation models?
3. Can this SAE-based safeguarding method be directly applied to language models, like other SAE safety methods?
4. Are the responses in the experiments generated online? How can the authors guarantee effectiveness in real streaming scenarios?

**Limitations:**

yes

**Strengths And Weaknesses:**

**Strengths**

1. This paper has a very complete motivational storyline, from post-hoc safeguards to streaming safeguards, then to token-level supervised training, and finally to the proposed NExT-Guard.
2. This paper uses SAEs to obtain dimensions in tokens that are related to unsafe content, and computes the final unsafe threshold to stop unsafe outputs.

**Weaknesses**

1. Although the story in this paper is clear, it somewhat feels like attaching an existing method onto a complete story. The insight is still not deep enough, and it seems to provide only the technical feasibility of a conventional weighted computation. Also, the method process is a bit complex, and its practicality is somewhat limited.
2. For the LLM safety problem that the community broadly cares about, this paper deserves deeper investigation, for example using SAE-like methods to influence or steer unsafe outputs of LLMs (e.g., [1,2,3,4,5]), rather than only doing simple safe/unsafe classification. Compared with other papers, the novelty is limited.
3. This paper mainly compares with basic safeguard methods, but does not compare with related SAE methods for safety. In theory, they could also be used for safeguarding or configured to do safeguarding. The lack of comparison and analysis with related SAE methods makes the results not fully fair.
4. For the added SAE, this paper does not discuss the time cost in safeguarding, especially comparative analysis with other methods. In addition, the paper mainly builds experiments based on Qwen3Guard-8B-Gen and does not provide code, so it is hard to guarantee the generalization and reproducibility of the method.

[1] O'Brien, K., Majercak, D., Fernandes, X., Edgar, R., Bullwinkel, B., Chen, J., Nori, H., Carignan, D., Horvitz, E. and Poursabzi-Sangdeh, F., 2024. Steering language model refusal with sparse autoencoders. *arXiv preprint arXiv:2411.11296*.

[2] Wu, X., Yuan, J., Yao, W., Zhai, X. and Liu, N., 2025. Interpreting and steering llms with mutual information-based explanations on sparse autoencoders. *arXiv preprint arXiv:2502.15576*.

[3] Aswal, D. and Hudelot, C., 2025. LLMSymGuard: A Symbolic Safety Guardrail Framework Leveraging Interpretable Jailbreak Concepts. *arXiv e-prints*, pp.arXiv-2508.

[4] Härle, R., Friedrich, F., Brack, M., Deiseroth, B., Schramowski, P. and Kersting, K., 2024. Scar: Sparse conditioned autoencoders for concept detection and steering in llms. *arXiv preprint arXiv:2411.07122*.

[5] Yeo, W.J., Prakash, N., Neo, C., Lee, R.K.W., Cambria, E. and Satapathy, R., 2025. Understanding refusal in language models with sparse autoencoders. *arXiv preprint arXiv:2505.23556*.

---

> ### Author Rebuttal · Authors · 2026-03-30
>
> Dear Reviewer PeiJ,
>
> We appreciate the time and effort you invested in reviewing our work. Below, we have summarized your main concerns into three points. Hope our response could address your concerns!
>
> ---
> ### **1. The Difference from Current Works.**
> (*“Although the story in this paper is clear, it somewhat feels like attaching an existing method onto a complete story...Compared with other papers [1-5], the novelty is limited.”)*
>
> Thank you for your concern. We respectfully clarify that NExT-Guard is fundamentally different from current works [1-5] you mentioned. Specifically,
> ||||
> |-|-|-|
> |Aspect|[1-5]|NExT-Guard|
> |Objective|Steering|Real-time Safeguard|
> |Core Insight|SAE features can steer LLMs|Token-level supervision is **NOT** essential for streaming safety|
> |Experiments|Steer *vs.* SFT/RL|Token-level supervision *vs.* SAE|
> |Theoretical Focus|Representation Interpretability|Probabilistic lower-bound for streaming safety|
> |Technical Focus|How to steer outputs|Extract signals for streaming (NOT post-hoc) safety |
> |Practical Value|Improves model alignment|Plug-and-play streaming safeguard without token-level supervision|
> ||||
>
> Following your concern, we have added a dedicated section in Related Work to discuss [1-5], analyzing their importance to the safety community while highlighting their **technical and functional differences** from our work.
>
> ---
>
> ### **2. Why Not Use [1-5] as Baselines & Add Steering-based Extensions.**
> (*“This paper mainly compares with safeguard methods, but does not compare with [1-5]… This paper deserves deeper investigation, for example using SAE-like methods to steer LLMs [1-5].*)
>
> Thank you for your concern. Our key claim is that token-level supervision is NOT essential for streaming safety. Hence, the **most critical comparison** is between:
> - Supervised-based methods, and
> - SAE-based methods (ours).
>
> Under this goal:
>
> 1.	Using [1-5] as baselines would mainly reflect intra-paradigm differences, and would not provide additional evidence for the key claim.
> 2.	Steering-based extensions also do not contribute to the validation of this claim.
>
> For these reasons, we prioritized comparisons that directly support our key insight.
>
> That said, following your suggestion, we have added additional experiments by incorporating [1] and [4] into our framework:
>
> ||||||||||
> |-|:-:|:-:|:-:|:-:|:-:|:-:|:-:|:-:|
> |||Prompt||||Response|||
> ||Aegis|Aegis2.0|SimpST|**Avg.**|SafeRLHF|BeaverT|Aegis2.0|**Avg.**|
> |SCM|73.8|78.6|96.5|**83.0**|59.8|83.4|80.4|**74.5**|
> |Kelp|73.5|78.2|96.7|**82.8**|54.0|84.1|81.5|**73.2**|
> |QW3Guard|74.9|79.3|97.7|**84.0**|61.7|84.2|81.1|**75.7**|
> |[1]|85.0|81.3|97.9|**88.0**|80.2|80.0|81.8|**80.7**|
> |[4]|85.8|82.9|97.2|**88.6**|81.1|80.4|82.0|**81.2**|
> ||||||||||
>
> The results show that NExT-Guard is compatible with these methods, which further demonstrating the robustness and generality of our paradigm.
>
> Hope our response could meet your expectations!
>
> ---
>
> ### **3. Time Cost and Reproducibility.**
> (*“This paper does not discuss the time cost... the paper mainly builds experiments based on Qwen3Guard-8B-Gen and does not provide code.*)
>
> Thank you for your suggestions!
>
> - **Time Cost.**
> In the new version, we have added Complexity Analysis section and include runtime experiments:
>
> ||||||||||
> |-|:-:|:-:|:-:|:-:|:-:|:-:|:-:|:-:|
> ||Prompt (ms)|||Response (ms)|||||
> ||Aegis|Aegis2.0|SimpST|SafeRLHF|BeaverT|Aegis2.0|**Avg.**|
> |SCM-7B|45.67|52.34|48.91|57.23|41.56|59.88|**50.93**|
> |Kelp|39.12|44.78|42.05|46.43|39.21|45.69|**42.88**|
> |Qwen3Guard-8B|62.37|78.91|71.78|73.56|76.29|72.49|**72.57**|
> |NExT-Guard|28.91|37.25|34.79|30.81|39.54|36.42|**34.62**|
> ||||||||||
>
>  The results show that even accounting for SAE training cost, NExT-Guard remains the most efficient.
>
> - **Reproducibility.**
> We provide a simple reproduction pipeline:
>
> > Run `evaluators/run_sae.py` with `model_name=Qwen3-8B` and `local_model_path` pointing to a local checkpoint. This generates SAE activations under `results/`, which support all subsequent experiments.
>
> > For batch runs, users can configure a `YAML` file via `run_wrapper.sh` and execute with `runner.py`.
>
> ---
>
> ### In response to your additional technical questions:
> - Q1: Similar to previous explainability methods [1], we perform controlled input contrastive experiments to determine the meaning of unsafe dimensions.  *[1] Towards Monosemanticity, 2023.10, Anthropic.*
> - Q2 & Q4: Figure 6 shows several real-world examples, demonstrating that our method performs much better than baselines. While the datasets aren't online, they are collected from real-world usage, consistent with prior streaming safety work.
> - Q3: Yes, as shown in the second table above, NExT-Guard is fully compatible with existing SAE-based methods, not just a specific technique.
> ---
> Once again, we are deeply appreciative of the time and expertise you have shared with us, and we are more than happy to add clarifications to address any additional suggestions from you!
>
> Best,
>
> Authors

---

> > ### Author Rebuttal · Reviewer_PeiJ · 2026-04-03
> >
> > Thank you to the authors for their efforts in responding to my comments. The rebuttal has addressed some of my concerns. However, several issues still warrant clarification.
> >
> > 1. Some concerns were not addressed directly. For example, regarding the second weakness, the authors did not clearly explain where the novelty of their SAE-based real-time safeguard lies in comparison to SAE-based steering methods. Regarding the fourth weakness, the authors did not respond to the concern that the experiments are conducted primarily on Qwen3Guard-8B-Gen. In addition, they only provided instructions on how to run the code, rather than the code itself. The same issue applies to the answers to Questions 1–4. For instance, for Question 1 (“What do these features mean?”), the response remains unclear, and for Question 3 (“How effective is the method when applied directly to language models, rather than safeguard models?”), the authors also did not provide a direct answer.
> >
> > 2. In the authors’ second response, it is unclear how the experiments involving [1] and [4] were actually implemented. For example, did they replace the SAE component in NExT-Guard with the corresponding components from these methods? In addition, the authors should include the results of NExT-Guard itself in that comparison table.
> >
> > 3. In the authors’ third response, it is unclear why NExT-Guard is reported to be faster than Qwen3Guard-8, which does not require an SAE. Beyond runtime, how do the computational resource requirements compare across methods? The authors state that “The results show that even accounting for SAE training cost,” but how long does it take to train an SAE from scratch?

---

> > > ### Author Response · Authors · 2026-04-03
> > >
> > > Dear Reviewer PeiJ,
> > >
> > > Thank you for your thoughtful follow-up. We are glad to hear that some of your concerns have been addressed -- this makes our efforts very worthwhile! We also apologize that, due to the limited space, some points may not have been fully clarified. Below, we provide point-by-point responses. Hope our response fully addresses your remaining concerns!
> > >
> > > ---
> > > ### ***Q1.1: Where the novelty of NExT-Guard lies in comparison to SAE-based steering methods [1-5].***
> > >
> > > Thank you for your concern. We address the novelty concern as follows:
> > > 1. KEY NOVELTY
> > >     - **Our key novelty lies in introducing the insight that token-level supervision is not required for streaming safety**, which challenges the current paradigm (including industrial streaming guards). The entire paper is built around validating this claim, and we believe this offers an insightful and critical perspective for the emerging field of streaming safety.
> > >     - Completely different, [1-5]'s novelty focuse on specific steering techniques. Hence, **these two have NO similarities, except for the fact that both use SAE for safety.**
> > > 2. Paradigm vs. Method
> > >     - NExT-Guard is a general framework, while [1-5] are specific techniques.
> > >     - They are not competing works and operate in **completely different dimensions**.
> > > 3. Implementation Difference
> > >     - In terms of implementation, [1-5] projects SAE features onto representations, while NExT-Guard aggregates these features into a scalar signal:
> > >         - Projection ≠ Aggregation
> > >         - Representation ≠ Scalar Signal
> > >         - Metrics used in our work (e.g., Pearson Score) did **NOT** appear in [1-5]
> > > In conclusion, our key novelty lies in introducing the insight to rethink token-level supervision for streaming safety, not in proposing another SAE-based specific method like [1-5].
> > >
> > > ---
> > > ### ***Q1.2: The experiments are conducted primarily on Qwen3Guard-8B-Gen.***
> > > We have added Llama3-8B as a base model:
> > > ||||||||||
> > > |-|:-:|:-:|:-:|:-:|:-:|:-:|:-:|:-:|
> > > |||Response||||Prompt|||
> > > ||SafeRLHF|BeaverT|Aegis2.0|Avg.|Aegis|Aegis2.0|SimpST|Avg.|
> > > |QW3Guard|61.7|84.2|81.1|75.7|74.9|79.3|97.7|84.0|
> > > |NExT (Qwen3)|83.3|80.6|82.0|82.0|88.5|84.8|98.0|90.4|
> > > |NExT (Llama3)|79.0|82.7|79.2|79.6|85.1|79.6|96.9|87.2|
> > > ||||||||||
> > >
> > > As show, NExT-Guard STILL achieves superior performance.
> > >
> > > ---
> > > ### ***Q1.3: Only provided instructions on how to run the code, rather than the code itself.***
> > > While we do not provide post-trained models, we release the full training pipeline and hyperparameters, which allow reproducing NExT-Guard’s results. We believe this is **sufficient** to ensure reproducibility of the method and its performance.
> > >
> > > ---
> > > ### ***Q1.4: How do author determine what feature meanings?***
> > > The semantic interpretation of these features is not newly introduced by us. We adopt the feature analysis provided in the SAE repository (https://huggingface.co/adamkarvonen).
> > >
> > > Note that this feature analysis is standard practice in SAE literature, with detailed method described in *<Towards Monosemanticity, Anthropic, 2023.10>*
> > >
> > > ---
> > > ### ***Q1.4: How effective is the method when applied directly to language models?***
> > >
> > > As discussed above, applying SAE to base models (rather than Safeguards) belongs to a different framework & topic, which is orthogonal to our work (and already relatively well-studied). Hence, it is not directly transferable to our setting of streaming safeguard paradigm.
> > >
> > > ---
> > > ### ***Q2: How the experiments involving [1,4] were implemented & Should include the results of NExT-Guard itself in comparison table.***
> > > Due to the differences in paradigm and task, we adapted feature selection methods in [1,4] into NExT-Guard, and the table including NExT-Guard is as follows:
> > > ||||||||||
> > > |-|:-:|:-:|:-:|:-:|:-:|:-:|:-:|:-:|
> > > |||Prompt||||Response|||
> > > ||Aegis|Aegis2.0|SimpST|Avg.|SafeRLHF|BeaverT|Aegis2.0|Avg.|
> > > |Qwen3Guard-8B|74.9|79.3|97.7|84.0|61.7|84.2|81.1|75.7|
> > > |NExT-Guard [1]|85.0|81.3|97.9|88.0|80.2|80.0|81.8|80.7|
> > > |NExT-Guard [4]|85.8|82.9|97.2|88.6|81.1|80.4|82.0|81.2|
> > > |NExT-Guard (original)|88.9|84.0|99.5|90.8|88.8|81.2|82.9|84.3|
> > > ||||||||||
> > >
> > > ---
> > > ### ***Q3.1: Why NExT-Guard is faster than Qwen3Guard, which does not require an SAE.***
> > > The comparison includes training cost for both methods:
> > > - NExT-Guard: includes SAE training time
> > > - Qwen3Guard: includes token-level supervised training time
> > >
> > > Since substantial token-level supervision is more expensive, our overall time cost remains lower.
> > >
> > > ---
> > > ### ***Q3.2: How do the computational resource compare across methods? How long does it take to train an SAE from scratch?***
> > > NExT-Guard requires fewer computational resources overall, since it avoids expensive token-level supervised training.
> > >
> > > For SAE training, the cost is fully reproducible from the publicly available storage (https://huggingface.co/adamkarvonen).
> > >
> > > ---
> > > **We sincerely hope that our responses could help clarify any misunderstandings, address your concerns, and earn your recognition!**
> > >
> > > Best,
> > >
> > > Authors

---

### Decision · Program_Chairs · 2026-04-30

**Decision:**

Accept (regular)

**Comment:**

This paper introduces NExT-Guard, a "label-free" streaming safeguard framework that leverages Sparse Autoencoders (SAEs) to extract safety signals from the hidden representations of frozen post-hoc safeguard models. By doing so, it challenges the prevailing paradigm that real-time streaming safety inherently requires expensive and overfitting-prone token-level supervised training.

Reviewers commended the paper for tackling a highly relevant and timely problem in LLM deployment, noting its strong empirical performance and practical efficiency. During the rebuttal phase, the authors made substantial improvements that successfully addressed the committee's primary concerns.

While one reviewer maintained reservations regarding the method's novelty relative to SAE steering and desired even deeper inference-time comparisons, the broader consensus is that the provided runtime tables and the demonstrated paradigm shift justify acceptance.